# 8q24 genetic variation and comprehensive haplotypes altering familial risk of prostate cancer

William D. Dupont[1,10], Joan P. Breyer[2,3,10], W. Dale Plummer[1], Sam S. Chang[4], Michael S. Cookson[5], Joseph A. Smith[4], University of Washington Center for Mendelian Genomics*, Elizabeth E. Blue [6], Michael J. Bamshad[7] & Jeffrey R. Smith[2,3✉]

The 8q24 genomic locus is tied to the origin of numerous cancers. We investigate its contribution to hereditary prostate cancer (HPC) in independent study populations of the Nashville Familial Prostate Cancer Study and International Consortium for Prostate Cancer Genetics (combined: 2,836 HPC cases, 2,206 controls of European ancestry). Here we report 433 variants concordantly associated with HPC in both study populations, accounting for 9% of heritability and modifying age of diagnosis as well as aggressiveness; 183 reach genome-wide significance. The variants comprehensively distinguish independent risk-altering haplotypes overlapping the 648 kb locus (three protective, and four risk (peak odds ratios: 1.5, 4, 5, and 22)). Sequence of the near-Mendelian haplotype reveals eleven causal mutation candidates. We introduce a linkage disequilibrium-based algorithm discerning eight independent sentinel variants, carrying considerable risk prediction ability (AUC = 0.625) for a single locus. These findings elucidate 8q24 locus structure and correlates for clinical prediction of prostate cancer risk.

[1] Department of Biostatistics, Vanderbilt University Medical Center, 2525 West End Avenue, Nashville, TN 37203, USA. [2] Department of Medicine, Division of Genetic Medicine, Vanderbilt-Ingram Cancer Center, and Vanderbilt Genetics Institute, Vanderbilt University Medical Center, 507 Light Hall, 2215 Garland Avenue, Nashville, TN 37232, USA. [3] Medical Research Service, Tennessee Valley Healthcare System, Veterans Administration, 1310 24th Avenue South, Nashville, TN 37212, USA. [4] Department of Urology, Vanderbilt University Medical Center, A-1302 Medical Center North, 1161 21st Avenue South, Nashville, TN 37232, USA. [5] Department of Urology, University of Oklahoma Health Sciences Center, Suite 3150, 920 SL Young Boulevard, Oklahoma City, OK 73104, USA. [6] Department of Medicine, Division of Medical Genetics, University of Washington, HSB H132, Seattle, WA 98195, USA. [7] Department of Pediatrics, Division of Genetic Medicine, and Center for Mendelian Genomics, University of Washington, HSB RR349, 1959 NE Pacific Street, Seattle, WA 98195, USA. [10] These authors contributed equally: William D. Dupont, Joan P. Breyer. *A list of authors and their affiliations appears at the end of the paper. ✉email: jeffrey.smith@vumc.org

Prostate cancer is the most common non-skin cancer in men. Twin studies estimate that prostate cancer heritability is twice that of breast cancer, at roughly 58%[1–3]. Segregation analyses indicate that its familial clustering best fits heritable models, but without clear consensus of mode of transmission[4–9]. The heritability of prostate cancer has proven more complex to decipher than that of other common cancers. The observation of familial clustering of breast cancer and subsequent identification of heritable risk attributable to *BRCA1* and *BRCA2* has had broad impact on risk prediction, precision care, and etiological understanding. Observations of familial clustering of prostate cancer led to analogous investigation by linkage[10]. But while 23 extended pedigrees were sufficient to detect the *BRCA1* locus[11], investigation of far greater numbers of pedigrees meeting criteria for hereditary prostate cancer[12] (HPC) yielded less prominent loci[13] and greater discordance across global study populations, underscoring its greater complexity. HPC pedigrees have three or more affected first-degree or second-degree relatives, and can manifest a relatively early age of diagnosis[4,12]. Collaborative efforts identified *HOXB13* as a causal gene[14]. But the *HOXB13* G84E mutation segregates in only 4% of HPC families, and only half of affected men in these families are carriers, illustrating causal heterogeneity even within a pedigree[15]. The etiologic spectrum could include a burden of numerous common low-risk polymorphisms, less common intermediate-risk variants, or diverse and rare high-risk mutations. A given family may segregate any combination of these.

To identify underlying genetic risk factors, we used family history as a proxy for genetic burden, comparing case probands from independent HPC pedigrees to controls. This design increases power to detect less common and stronger risk variants, including large-effect mutations such as at *HOXB13*[16,17]. Here we present results of a comprehensive investigation of the 8q24 locus, outlined in Fig. 1. This is a region with high prior probability of impacting familial prostate cancer risk. The first genome-wide association study (GWAS) of prostate cancer detected a significant association at 8q24[18], and colon, breast, ovarian, bladder, pancreatic, and hematologic cancers are also associated with this locus[18–23]. An infrequent 8q24 variant carrying strong risk for prostate cancer was also detected in a deCODE study of Icelanders[21], and confirmed by the International Consortium for Prostate Cancer Genetics (ICPCG)[22]. Numerous cataloged GWAS SNPs for prostate cancer reside at 8q24[18–27]. They fall into distinct linkage disequilibrium (LD) blocks, suggesting the presence of multiple underlying functional variants.

This study seeks to comprehensively identify familial prostate cancer risk variants at 8q24 that are concordantly observed within independent sets of study subjects, to identify their corresponding ancestral haplotypes, and variants best detecting the independent risk signals on each haplotype. We employ array-based genotype detection followed by imputation against reference genomes of the Haplotype Reference Consortium (HRC)[28], enabling detection of relatively rare risk variants that we confirm by orthogonal assay. We discover multiple variants independently and multiplicatively associated with both risk and protective effects at genome-wide significance. Subsets of these modify both age of diagnosis and disease aggressiveness. In order to resolve the distinct underlying risk-altering signals, we reconstruct ancestral risk-altering haplotypes, and introduce an algorithm systematically employing LD patterns to identify the individual variants that best detect the risk-altering signal of each of these haplotypes. We sequence carriers of a near-Mendelian risk haplotype to delineate causal mutation candidates. We estimate the overall contribution of the 8q24 locus to prostate cancer heritability, and its capacity for clinical risk prediction.

## Results

**8q24 variants associated with hereditary prostate cancer.** We identified candidate risk variants within HPC cases and controls of the ICPCG, aggregated from 12 global studies of European ancestry subjects (Table 1). We clustered array-based data de novo for optimal calling of rare variants, and imputed from reference whole genome sequence of the HRC[28]. Under additive

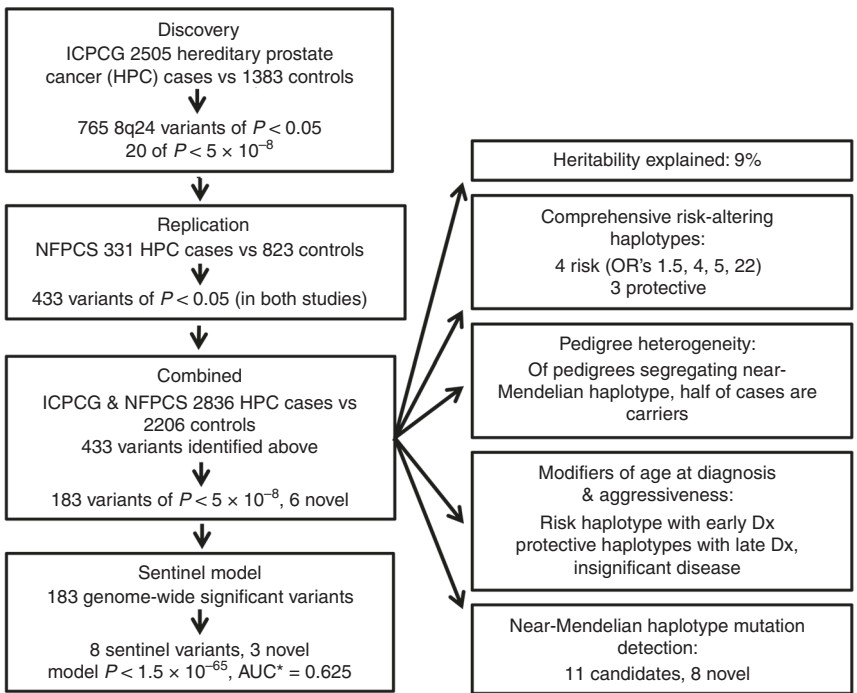

**Fig. 1 Study overview.** HPC hereditary prostate cancer, ICPCG International Consortium for Prostate Cancer Genetics, NFPCS Nashville Familial Prostate Cancer Study, Dx diagnosis. *Shrunken area under the receiver operator characteristic curve (AUC).

| | NFPCS | | | ICPCG[a] | | |
|---|---|---|---|---|---|---|
| | Controls | HPC cases | FPC cases | Controls | HPC cases | FPC cases |
| European ancestry, count | 823 | 331 | 343 | 1383 | 2505 | 2 |
| Mean age at Dx or screen | 62 | 60 | 56 | nr | 60 | 54 |
| ≤55 years | 23% | 28% | 43% | nr | 26% | 50% |
| 56–65 years | 40% | 47% | 56% | nr | 46% | 50% |
| ≥66 years | 37% | 25% | 1% | nr | 27% | 0% |

**Table 1 Study populations.**

[a]Subjects of 12 aggregated studies (dbGaP phs000733.v1.p1). An HPC case is a proband from a pedigree with ≥3 affected men. An FPC case is a proband from a pedigree with only two affected men. All case and control subjects are unrelated. HPC cases were used for all analyses, while FPC cases were evaluated in Supplementary Fig. 1 and Supplementary Data File 1.
NFPCS Nashville Familial Prostate Cancer Study, ICPCG International Consortium for Prostate Cancer Genetics, HPC hereditary prostate cancer, FPC familial prostate cancer, Dx diagnosis, nr not recorded.

genetic models, twenty 8q24 variants were associated with HPC at genome-wide significance (Supplementary Fig. 1 and Supplementary Data File 1). The least frequent of these had a risk allele frequency of 0.2% among controls, indicating that the strategy was capable of detecting rare risk variants. A 648 kb 8q24 interval was distinguished by 765 variants of $P \leq 0.05$ (Supplementary Data File 1); the interval harbors non-coding genes, the *POU5F1B* retrogene[29], and regulatory elements. All 28 previously identified 8q24 prostate cancer GWAS SNPs among subjects of European ancestry were within this variant set.

We sought to replicate observations in independent European ancestry subjects from the Nashville Familial Prostate Cancer Study (NFPCS, Table 1). The NFPCS is a study of HPC case probands and screened controls without a personal or family history of prostate cancer. The NFPCS (1497 subjects) was a third the size of the collective ICPCG study populations (3890 subjects). The mean age of HPC case diagnosis was consistent with the earlier age of onset of familial disease[4]. We array-genotyped NFPCS subjects with de novo clustering as a basis for imputation against the HRC reference. Of the 765 variants at nominal significance in ICPCG subjects in this interval, 743 were either directly genotyped in the NFPCS or had an imputation $R^2 \geq 0.8$, while the remaining 22 had an $R^2 \geq 0.5$. In total, 433 of the 765 (57%) were concordantly nominally significant in the NFPCS under additive models ($P \leq 0.05$, Supplementary Fig. 1 and Supplementary Data File 1). In the NFPCS, false discovery rates for 200 variants were below 0.005, and 100 variants were significant after Bonferroni correction. We genotyped 61 of the 765 variants by orthogonal assays in the NFPCS, confirming association for 54 of them (Supplementary Data File 1). None of the 433 variants reached genome-wide significance in the smaller NFPCS HPC study population alone. We had recruited a separate series of 343 additional NFPCS cases with an age of diagnosis under 66 years and with only one additional affected relative ("FPC" cases). In combined data of the NFPCS HPC and FPC cases, 63 variants met genome-wide significance (Supplementary Fig. 1 and Supplementary Data File 1).

In the combined ICPCG and NFPCS HPC case and control subjects, 183 of the concordant variants met genome-wide significance (Fig. 2 and Supplementary Data File 1); six remained significant when concurrently adjusted for the previously known GWAS variants[18–27] (Table 2). Forty-nine additional replicating variants that did not reach genome-wide significance among combined subjects also remained significant after adjustment for known GWAS variants (Supplementary Data File 1). Minor alleles may be partitioned by effect size: 8 of odds ratios (OR's) > 4; 68 of OR between 2 and 4; 43 of OR between 1.5 and 2; 157 of OR between 1 and 1.5; and 157 of OR < 1 (Fig. 2 and Supplementary Data File 1). Effect sizes within the range known for breast cancer predisposition by pathogenic variants of *BRCA1* and *BRCA2* (ORs > 8)[30] were observed at rs182352457_*A* ($P = 8 \times 10^{-9}$,

OR = 8.4), rs188140481_*A* ($P = 2 \times 10^{-12}$, OR = 8.5), and rs138042437_*G* ($P = 3 \times 10^{-12}$, OR = 9.1).

**Numerous associated variants reside on few risk-altering haplotypes.** Disease-associated alleles are inherited in the context of ancestral haplotypes. Alleles that are unique to (that "mark") the DNA segment on which a causal mutation is introduced will evidence association with disease. Recombination can diminish the correlation between a causal mutation and these alleles with time. We sought to understand how the associated alleles are transmitted as haplotypes, shedding light upon complex correlations. As discussed further below, we also identified which variant best detects the risk-altering signal(s) carried by each of these haplotypes. We conducted haplotype-based tests of association in HPC subjects of the combined study populations by sliding-window analyses of all 433 concordantly-associated variants[29,31]. As an overview, each haplotype within a specified window of $N$ variants was tested for association with disease under an additive model, and windows of variable width were moved incrementally along the map with sequential tests. Where combinations of alleles mark a given haplotype and can be uniquely aligned as the window slides along the map, a model of a distinct ancestral risk-altering haplotype can be built. We required concordant association of a given observed haplotype in the NFPCS and the ICPCG ($P \leq 0.05$ in each, separately). We detected one rare, high-risk haplotype (approaching Mendelian effect size), an infrequent moderate-risk haplotype, a common low-risk haplotype, a rare moderate-risk haplotype, and three protective haplotypes. A protective haplotype is one in excess among controls relative to cases. These are designated haplotypes A through G in Fig. 3. Nearly all of the 433 variants had a minor allele that was carried specifically by only one among the risk-altering haplotypes; the direction of effect of the minor allele was the same as that of its corresponding haplotype (Fig. 3 and Supplementary Fig. 2). Each of these haplotypes was consistently and redundantly detected by sliding window analyses.

For sliding windows of up to 20 adjacent variants, the windows of peak effect size and of peak significance for each haplotype are illustrated in Fig. 3 (for each of haplotypes D, F, and G this is a single window). The window of peak effect size on high-risk haplotype A spanned 19 variants from rs138042437_*G* to rs13251915_*T* ($P = 2 \times 10^{-9}$, OR = 12.6). Among the various study populations, this haplotype was at highest frequency in Finns, carried by 8.9% of the cases, vs. 3.9% of all HPC cases and 0.3% of controls. Among individual variants marking haplotype A, rs138042437_*G* had the largest effect size ($P = 3 \times 10^{-12}$, OR = 9.1), while rs7832031_*A* was most significant ($P = 5 \times 10^{-13}$, OR = 1.6). The minor allele of the first prostate cancer GWAS SNP to have been discovered[24], rs1447295_*A* is present on the right (telomeric) flank of haplotype A. The window of peak effect

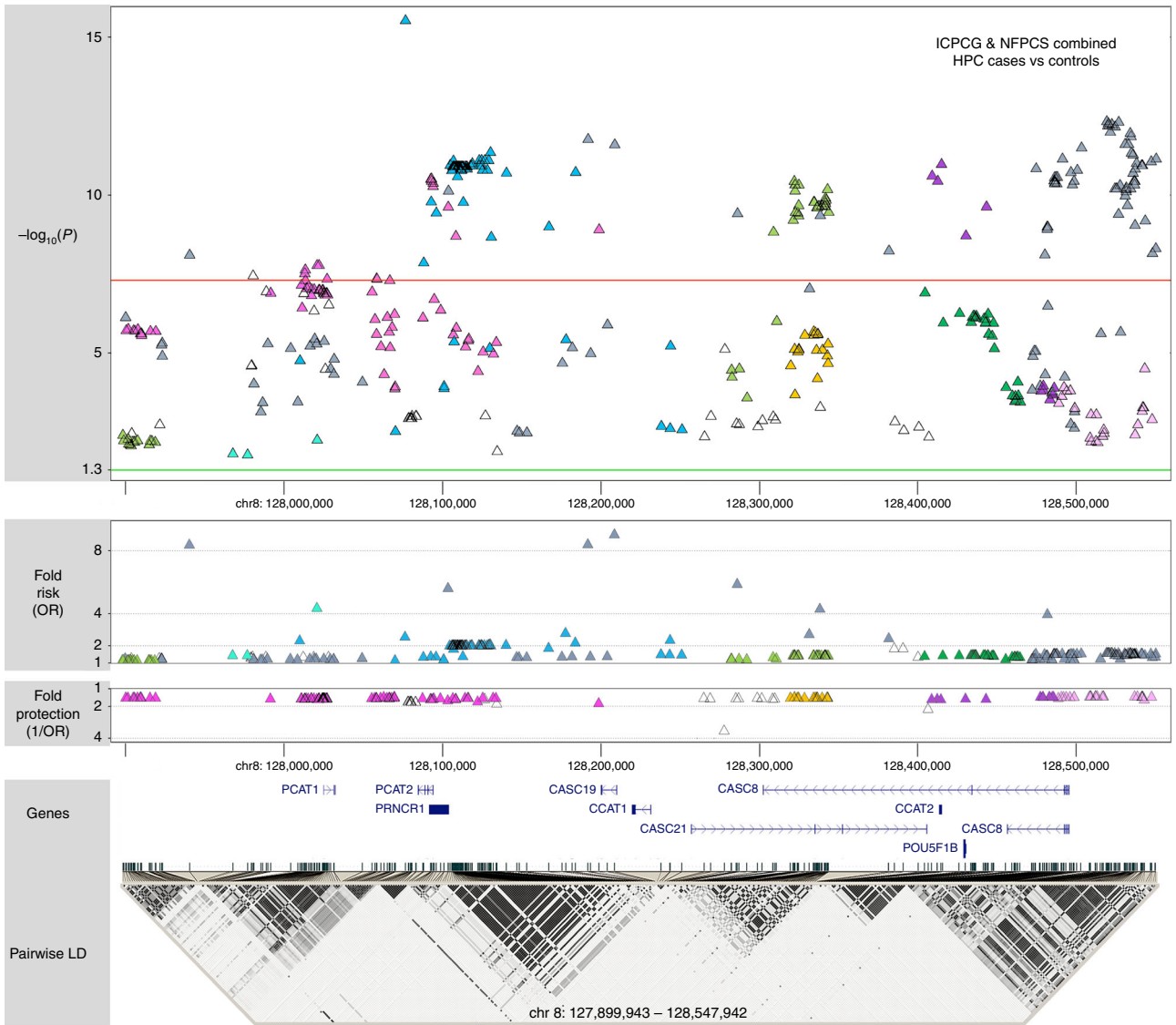

**Fig. 2 Association of 433 concordantly significant 8q24 genetic variants with HPC in combined NFPCS and ICPCG subjects.** Each variant is color-coded to delineate the corresponding risk-altering haplotype marked by its minor allele (haplotypes are separately depicted in Fig. 3 and Supplementary Fig. 2). At top is a Manhattan plot of the combined study populations depicting variant positions on the x-axis and $-\log_{10} P$ values on the y-axis. The horizontal red line corresponds to the genome-wide significance threshold of $P = 5 \times 10^{-8}$ to accommodate multiple comparisons; the green line to $P = 0.05$. Each data point depicts the result of a multiplicative logistic regression model (additive genetic model), with two-sided significance assessed using Wald tests. The middle plot displays corresponding effect sizes, either risk or protective. At bottom is a map of regional genes (UCSC hg19), and a pairwise LD matrix depicting $R^2$ values among HPC cases of the combined studies. Source data are provided in Source Data file tab 1.

---

**Table 2 Genome-wide significant variants associated with HPC among combined subjects, remaining significant under adjustment for previously known variants.**

| rsID_allele | hg19 chr8 position | $P$ | OR | 95% CI | $P_{adj}$[a] | $OR_{adj}$ | 95% $CI_{adj}$ |
|---|---|---|---|---|---|---|---|
| rs9297750_G | 128022973 | $1.7 \times 10^{-8}$ | 0.77 | 0.70–0.84 | 0.010 | 0.77 | 0.63–0.94 |
| rs10956349_C | 128059066 | $4.5 \times 10^{-8}$ | 0.80 | 0.74–0.86 | 0.036 | 0.89 | 0.79–0.99 |
| rs10956350_T | 128059283 | $4.6 \times 10^{-8}$ | 0.80 | 0.74–0.86 | 0.034 | 0.88 | 0.79–0.99 |
| rs4288339_C | 128067300 | $5.1 \times 10^{-8}$ | 0.80 | 0.74–0.87 | 0.038 | 0.89 | 0.79–0.99 |
| rs12678349_T | 128198564 | $1.3 \times 10^{-9}$ | 0.64 | 0.56–0.74 | $1.9 \times 10^{-5}$ | 0.72 | 0.62–0.84 |
| rs4871790_C | 128441535 | $2.4 \times 10^{-10}$ | 0.77 | 0.72–0.84 | 0.037 | 0.89 | 0.79–0.99 |

[a]Each variant remains significant in a multivariable logistic regression model adjusted concurrently for all known 8q24 prostate cancer GWAS SNPs (rs77541621, rs4242384, rs4242382, rs11986220, rs188140481, rs138042437, rs16901979, rs6983267, rs12682344, rs6983561, rs10505477, rs1447295, rs12682374, rs4506170, rs183373024, rs16902104, rs1016343, rs56005245, rs16902094, rs445114, rs10086908, rs13252298, rs73351629, rs72725879, rs1914295, rs1487240, rs5013678, rs17464492, rs7812894, rs12549761, rs78511380, rs190257175). Forty-nine additional variants with $P < 0.05$ after adjustment for known variants are given in Supplementary Data File 1 (Tab 3). Two-sided significance was assessed using Wald tests, with the genome-wide significance threshold of $P < 5 \times 10^{-8}$.

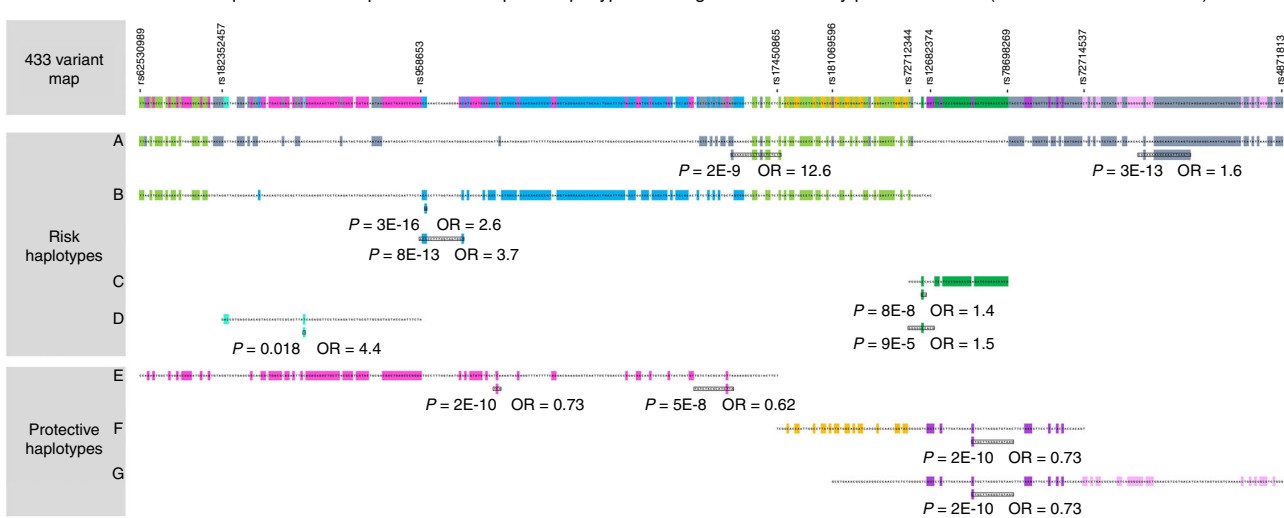

**Fig. 3 Seven HPC risk-altering haplotypes carry the minor alleles of the 8q24 prostate cancer risk variants.** These haplotypes are comprehensively reconstructed from all associated variants, displayed ordinally rather than on a physical scale. Four haplotypes conveying risk are illustrated above three others conveying protection. Each haplotype is distinguished from other risk-altering haplotypes by minor alleles of variants that are colorized (e.g., variants marking haplotype E are in magenta shade). Most variants had a minor allele residing upon only one of the risk-altering haplotypes. Variants in light green were the exception, with minor alleles that were shared by haplotypes A and B (potentially contributing to risk of both). The few remaining variants with minor alleles that were not unique to one among these haplotypes are left uncolorized. The direction of effect of a given minor allele was consistent with the direction of effect of the corresponding haplotype. Windows of peak significance and of peak effect size for each haplotype are denoted by inset boxes with corresponding P value and OR. Tests of association between hapotype and disease used multiplicative logistic regression models (additive genetic models), with two-sided significance assessed using Wald tests. For haplotypes D, F, and G, the window with the smallest P value also is that with the greatest effect size. Only near-Mendelian haplotype A extends across the full length of the map. Haplotypes are presented with additional detail in Supplementary Fig. 2. Source data are provided in Source Data file Tab 2. Variants at the ends of haplotypes A-G are denoted on the top map.

size of moderate-risk haplotype B spanned 17 variants from rs958653_A to rs10095770_C ($P = 8 \times 10^{-13}$, OR = 3.7) carried by 10.1% of cases and 3.0% of controls. This haplotype was also carried by 10.5% of the separate NFPCS FPC cases (FPC case vs. control $P = 5 \times 10^{-5}$, OR = 4.5). Among individual variants marking haplotype B, rs139046764_A had the largest effect size ($P = 4 \times 10^{-6}$, OR = 2.9), while rs77541621_A was most significant ($P = 3 \times 10^{-16}$, OR = 2.6). Low-risk haplotype C had a window of peak effect size spanning 10 variants from rs72712344_G to rs6983267_G ($P = 9 \times 10^{-5}$, OR = 1.5) carried by 11.7% of cases and 8.0% of controls. The individual variant of greatest significance marking haplotype C was rs10441525_C ($P = 1 \times 10^{-7}$, OR = 1.4). Haplotype D was a second potential rare and moderate-risk haplotype carried by 17 of 2836 cases and 3 of 2206 controls. The window of peak effect size of protective haplotype E spanned 15 variants from rs78512696_T to rs138042437_A ($P = 5 \times 10^{-8}$, OR = 0.62) carried by 8.7% of cases and 13.6% of controls. The individual variant of strongest effect marking haplotype E was rs12678349_T ($P = 1 \times 10^{-9}$, OR = 0.64), and that of greatest significance was rs5013678_C ($P = 2 \times 10^{-10}$, OR = 0.73). Protective haplotypes F and G overlap, sharing a mid-segment but with distinct flanks. A single window of peak effect size and of significance is shared by haplotypes F and G, spanning 16 variants from rs4871790_C to rs7825928_C ($P = 2 \times 10^{-10}$, OR = 0.73), carried by 48% of cases and 59% of controls. The minor allele of the second prostate cancer GWAS SNP to have been discovered[27], rs6983267_T, falls in this shared segment.

As an alternative approach, we considered that the effect of a given haplotype might be best detected by instead evaluating only the subset of variants with minor alleles specifically distinguishing it from the other risk-altering haplotypes. This required evaluation of a distinct set of variants for each respective risk-altering haplotype. The windows of peak effect size and of significance in this secondary analysis are presented in Supplementary Fig. 2, for each haplotype revealing a stronger effect at a window overlapping the previously observed one: for high-risk haplotype A, $P = 2 \times 10^{-5}$, OR = 22.3; for moderate-risk haplotype B, $P = 6 \times 10^{-6}$, OR = 5.1. These windows also encompassed the peak effect signals of the NFPCS and ICPCG when separately evaluated.

Multiple affected family members were available for 32 NFPCS HPC pedigrees. Three of these pedigrees segregated the high-risk haplotype A. However, the haplotype was carried by only six of the 13 affected men of the three pedigrees with DNA samples available for genotyping. The affected proband of one of the three pedigrees, who was diagnosed at age 43, instead carried haplotype B. This heterogeneity was reminiscent of pedigrees segregating the chromosome 17 *HOXB13* G84E mutation, where similarly only half of all affected men genotyped within the pedigrees were carriers[15]. Five pedigrees segregated haplotype B, including a homozygous case diagnosed at age 50.

**Modifiers of age at prostate cancer diagnosis and aggressiveness.** The relatively young age of diagnosis of HPC cases suggests that some variants might modify age of onset. Two variants of moderate-risk haplotype B were associated with an earlier age of diagnosis among HPC cases (rs7005144_A, $P = 0.023$, and rs191785584_G, $P = 0.019$), but nearly all other variants marking this haplotype also approached significance ($P < 0.1$). Among all combined (HPC and FPC) cases, 64 of the 77 variants distinguishing haplotype B were associated with a younger age of diagnosis ($P < 0.05$). The most significant were rs16901984_C and rs191785584 _G ($P = 0.0026$ for each, vs. $P = 0.0133$ for the correlated sentinel rs77541621_A discussed below). The mean age at diagnosis of carriers was 1.4 years younger than non-carriers (and 7 years younger than the US and UK average). Six variants

of protective haplotype G were associated with a later age of diagnosis among HPC cases (rs13250904_A, rs13251194_A, rs7819102_T, rs6981424_A, rs13260378_T, and rs75428928_C; P range 0.015–0.025). The mean age of diagnosis was 0.6 years older for carriers. This corroborates a prior observation that GWAS SNP rs6983267_T, also marking haplotype G, was associated with a later age of diagnosis[32].

Alleles carried by protective haplotype F were associated with prostate cancer severity, in excess among HPC cases with insignificant disease relative to either moderate or aggressive disease. Ten variants (each with an allele specific to the centromeric segment of haplotype F that is not shared with haplotype G) were significant in both comparisons: rs10107982_C, rs7838810_C, rs453875_A, rs622856_T, rs622853_A, rs620861_A, rs443053_T, rs587948_G, rs623401_G, and rs10956359_C. The most significant in comparison of aggressive HPC cases to insignificant HPC cases was rs10956359_C (P = 0.009, OR = 0.72); the most significant in comparison of moderate to insignificant cases were rs587948_G and rs623401_G (P = 0.006, OR = 0.73 for each). All ten were also significant in analogous comparisons of HPC and FPC cases combined. Carrier frequencies were not significantly different between aggressive and moderate cases.

**Contribution of 8q24 to HPC heritability.** We used variance components analysis by a restricted maximum likelihood approach to estimate the proportion of heritability explained by 8q24[33]. The estimated heritability ($h^2_{SNP}$) explained by the 765 nominally significant variants in ICPCG HPC subjects was 14.5% (SE = 0.028, P = $9 \times 10^{-47}$). In the combined ICPCG and NFPCS HPC subjects, the heritability explained by the 433 concordantly significant variants was 9.1% (SE = 0.023, P = $2 \times 10^{-61}$). In the NFPCS HPC subject set alone this heritability estimate was 12.4% (SE = 0.045, P = $5 \times 10^{-19}$). Under an alternative approach using Bayesian variable selection regression, the proportion of variance explained by the 433 variants in combined subjects was 7.3% (model $\log_{10}$(Bayes factor(BF)) = 158.5).

**Sentinel variants and relation to risk-altering haplotypes.** Sentinel variants are those best detecting independent risk-altering signals. We identified sentinel variants using a recursive algorithm that is based explicitly upon LD patterns (see RISSc sentinel algorithm in Methods). The algorithm identifies variants that optimally detect the risk signal of a given LD bin, and those which detect independent risk signals across LD bins under mutual adjustment. Because any given set of variants may be sufficiently correlated that they are not significant under mutual adjustment, the algorithm judiciously employs LD patterns to

ensure that variants optimally detecting independent risk signals are retained in the model, while others are filtered. The algorithm works well with highly correlated variants to yield a multivariable model of sentinels with low pairwise correlation coefficients and high significance under mutual adjustment.

We applied the algorithm to systematically identify sentinels among those at genome-wide significance, and assessed how they were organized relative to the risk-altering haplotypes identified above. This identified eight independent sentinels with a mutually-adjusted overall model P = $2 \times 10^{-65}$ (Table 3). Where more than one sentinel marks a given risk-altering haplotype, their joint inheritance will carry a compound effect. Two sentinels mark high-risk haplotype A, *rs188140481_A ($P_{adj} = 2 \times 10^{-9}$, $OR_{adj} = 6.3$) and rs7832031_A ($P_{adj} = 4 \times 10^{-8}$, $OR_{adj} = 1.4$). These reside within the two regions on sliding window haplotype analysis of peak effect size and significance, respectively. Moderate risk haplotype B is marked by sentinels *rs77541621_A ($P_{adj} = 3 \times 10^{-9}$, $OR_{adj} = 2.1$) and *rs1016343_T ($P_{adj} = 1 \times 10^{-5}$, $OR_{adj} = 1.3$). Sentinel rs74822356_G ($P_{adj} = 5 \times 10^{-5}$, $OR_{adj} = 1.3$) has a minor allele that marks both haplotypes A and B. The minor alleles of two sentinels mark protective haplotype E, again suggesting a compound effect: *rs1487240_G ($P_{adj} = 1 \times 10^{-7}$, $OR_{adj} = 0.77$) and rs12678349_T ($P_{adj} = 6 \times 10^{-7}$, $OR_{adj} = 0.69$). The lone final sentinel *rs6983267_T ($P_{adj} = 2 \times 10^{-7}$, $OR_{adj} = 0.81$) is one of the earliest GWAS SNPs identified, marking the shared segment of protective haplotypes F and G. Variants previously in the GWAS Catalog are denoted by an asterisk. No variants marking haplotype C were retained when adjusted for these sentinels. Pairwise LD among the sentinels is low ($R^2$ range 0–0.13). Sentinel rs12678349_T remains significant after adjustment for all known prostate cancer GWAS variants (Table 2), while sentinels rs7832031_A and rs74822356_G better detect their respective underlying risk signals than correlated, previously published variants. Pairwise interaction terms for the sentinels were not significant, supporting a multiplicative rather than synergistic model.

Using an alternative Bayesian approach[34], the variants of greatest posterior inclusion probability (PIP) on each risk-altering haplotype were: rs138042437_G on haplotype A ($\log_{10}$(BF) = 12.8, PIP = 0.65 (vs rs188140481_A with $\log_{10}$(BF) = 12.8, PIP = 0.57); rs77541621_A on haplotype B ($\log_{10}$(BF) = 14.1, PIP = 0.98); rs12678349_T on protective haplotype E ($\log_{10}$(BF) = 7.1, PIP = 0.99); and rs12682374_G on the shared segment of protective haplotypes F and G ($\log_{10}$(BF) = 7.1, PIP = 0.95 (vs rs6983267_T $\log_{10}$(BF) = 9.0, PIP = 0.37).

**Causal mutation candidates of high-risk haplotype A.** We sequenced seven cases representing a breadth of clinical features, each carrying large-effect haplotype A (Supplementary Data

**Table 3 Sentinel variants associated with HPC among combined subjects.**

| rsID_allele | hg19 chr8 position | Haplotype | Individual variants, unadjusted | | | RISSc sentinel set, mutually adjusted | | |
|---|---|---|---|---|---|---|---|---|
| | | | P | OR | 95% CI | P | OR | 95% CI |
| rs1487240_G | 128021752 | E | $1.7 \times 10^{-8}$ | 0.77 | 0.70–0.84 | $1.3 \times 10^{-7}$ | 0.77 | 0.70–0.85 |
| rs77541621_A | 128077146 | B | $3.0 \times 10^{-16}$ | 2.62 | 2.08–3.30 | $3.0 \times 10^{-9}$ | 2.10 | 1.64–2.68 |
| rs1016343_T | 128093297 | B | $1.7 \times 10^{-10}$ | 1.37 | 1.24–1.50 | $1.3 \times 10^{-5}$ | 1.26 | 1.14–1.40 |
| rs188140481_A | 128191672 | A | $1.7 \times 10^{-12}$ | 8.47 | 4.68–15.3 | $2.4 \times 10^{-9}$ | 6.27 | 3.43–11.5 |
| [a]rs12678349_T | 128198564 | E | $1.3 \times 10^{-9}$ | 0.64 | 0.56–0.74 | $6.0 \times 10^{-7}$ | 0.69 | 0.59–0.80 |
| [a][b]rs74822356_G | 128320976 | A,B | $3.7 \times 10^{-11}$ | 1.44 | 1.29–1.61 | $4.9 \times 10^{-5}$ | 1.26 | 1.13–1.41 |
| rs6983267_T | 128413305 | F,G | $1.1 \times 10^{-11}$ | 0.76 | 0.70–0.82 | $2.2 \times 10^{-7}$ | 0.81 | 0.74–0.88 |
| [a][b]rs7832031_A | 128516952 | A | $4.8 \times 10^{-13}$ | 1.58 | 1.39–1.78 | $4.0 \times 10^{-8}$ | 1.43 | 1.26–1.63 |

Results of multiplicative logistic regression models (additive genetic models) of each individual sentinel, as well as of the set under mutual adjustment. Two-sided significance was assessed using Wald tests. Each individual variant met genome-wide significance (P < $5 \times 10^{-8}$).
[a]Novel.
[b]Better detects the corresponding risk signal than previously known correlated variants.

**Table 4 Causal mutation candidates of high-risk haplotype A.**

| rsID_allele | hg19 chr8 position | Type | Reference population frequency[b] | | | | |
|---|---|---|---|---|---|---|---|
| | | | NFE | FIN | AFR | EAS | TOPMed |
| [a]rs1290265560_G | 127905152 | STR | | | | | 0.005 |
| [a]rs182352457_A | 127941793 | SNP | 0.007 | 0.016 | 0.001 | 0 | 0.003 |
| rs183373024_G | 128104117 | SNP | 0.008 | 0.016 | 0.001 | 0 | 0.004 |
| rs188140481_A | 128191672 | SNP | 0.006 | 0.015 | 0.001 | 0 | 0.003 |
| [a]rs1428102803_T | 128205878 | SNP | 0.007 | 0.016 | 0.001 | 0 | 0.005 |
| rs138042437_G | 128208369 | SNP | 0.006 | 0.016 | 0.001 | 0 | 0.003 |
| [a]rs201885483_G | 128285408 | indel | 0.005 | 0.014 | 0.001 | 0 | 0.003 |
| [a]rs201361304_TAC | 128337272 | indel | 0.033 | 0.078 | 0.006 | 0 | 0.005 |
| [a]rs779803467_A | 128337273 | STR | 0 | 0 | 0.014 | 0 | 0.001 |
| [a]rs1405065666 _T | 128420318 | STR | 0.003 | 0.007 | 0.002 | 0 | |
| [a]rs78311688_C | 128479976 | SNP | 0.006 | 0.016 | 0.001 | 0 | 0.003 |

[a]Novel.
[b]gnomAD NFE (non-Finnish European), FIN (Finnish), AFR (African American), EAS (East Asian) or TOPMed reference frequency. Supplemental Data File 2 provides variant detail.

File 2). We detected 2347 variants within the map interval. We inferred phase by identifying the shared haplotype A alleles, corroborated by reads spanning adjacent variants, which was fully consistent with the previously-defined haplotype A. We considered alleles on the disease haplotype and with rare reference population frequencies as potential causal mutation candidates. Our work above had observed alleles of peak effect size on haplotype A to be of a frequency ≤ 0.7% in gnomAD[35] and TOPMed[36] databases. The gnomAd reference provided frequency estimates for all but 40 of the detected variants. For these 40, either a TOPMed frequency was available or the minor allele was not specific to haplotype A.

We identified eleven candidate variants potentially causally related to haplotype A risk (Table 4). One of these cases was homozygous across an interval spanning chr8:128166556–128233992, including three of these candidates (rs188140481_A, rs1428102803_T, and rs138042437_G). We had evaluated seven of the eleven candidates for statistical association with disease; of these, the strongest risk effect was observed for rs138042437_G (Supplementary Data File 1). CADD scores do not indicate deleterious function for the identified mutational candidates (score range 0.04–3.47)[37]. However, functional annotations of this locus are incomplete, for example not currently encompassing alternative exons of POU5F1B[38]. rs138042437_G and rs188140481_A are within ENCODE-designated enhancers, and rs188140481_A is also within an alternative exon of the noncoding gene CASC19. rs183373024_G is within the non-coding gene PRNCR1, within an ENCODE-designated FOXA1 binding region, and within a HOXB13 binding region of the LNCaP and VCaP cell lines[39]. Multiple mutation candidates reside within introns of the non-coding genes CASC8, CASC19, CASC21, PCAT1, and POU5F1B[38].

Sequencing also identified a structural variant. Sequence originating from the second intron of CD96 on 3q13.13 was inserted at chr8:128533850–128533851. This translocation was absent from haplotype A and absent from the hg19 reference assembly, but was observed on each of the other sequenced chromosomes (Supplementary Data File 2). We also observed absence of the insertion on 7 of 60 pan-continental chromosomes of 1000 Genomes Phase 3 subjects with high-coverage sequence. Absence of the insertion is the minor allele, and is not rare. This structural variant likely underlies the described physical interaction between 8q24 and 3q13.13 in prostate adenocarcinoma cell lines PC-3, LNCaP, and DU-145 (none of which carry haplotype A)[40].

## Discussion

Familial clustering of prostate cancer motivates the search for germline risk variants to provide insight into disease origin and to guide hereditary cancer care. In order to detect infrequent and strong risk variants in hereditary prostate cancer families in the setting of complexities such as incomplete penetrance and genetic heterogeneity, our study design selectively compared independent cases with a family history of the disease to controls without. To our knowledge, the NFPCS was the first conducted under this design[41,42]. Enrichment of disease allele frequency among such cases coupled with imputation against a large reference panel proved capable of detecting rare but recurrent risk variants. Subsequent sequencing of carriers of an associated haplotype then identified causal mutation candidates. A limitation of this approach is that it would not detect rarer mutations, potentially private to a given family. Replication across independent study populations aided distinction of true from false positive observations. Replication lends confidence and serves as an effective filter against false positives, but also has potential for rejection of a true observation due to constrained power in either study population alone. Potential genetic heterogeneity across subjects from different global locations represents another potential limitation. Adjustment for genetic ancestry guards against confounding due to population structure, but a disease variant specific to one study population might have reduced power for detection and fail to replicate. Potential phenotypic heterogeneity is another limitation. The investigated cases represent a spectrum of age of diagnosis and of aggressiveness. Reducing phenotypic heterogeneity could constrain genetic heterogeneity to aid investigation, though with the competing consideration of fewer subjects in a given stratum. Power to detect a variant of strong risk effect would be increased by enrichment among cases with a strong family history of the disease. An early age of diagnosis is also a recognized clinical facet among familial cases. Measured effect sizes could be larger under a familial case–control study, by virtue of disease allele enrichment, than would be expected under a case–control study unselected for family history. While family history criteria for HPC were uniform across sites, subsets of ICPCG cases were also further selected based upon early age of diagnosis or aggressiveness. Power to detect a risk variant predisposing selectively to aggressive prostate cancer might be reduced by inclusion of less aggressive cases; conversely, power to detect variants that protect from aggressive prostate cancer might be increased. Intriguingly, haplotypes that were protective in this

study were also associated with a later age at diagnosis and with less severe disease among cases.

We identified 183 variants associated with HPC at genome-wide significance, six of them not previously reported. Four variants at genome-wide significance had odds ratios as great as 9, while 70 variants had odds ratios over 2. The numerous risk-altering variants are complexly correlated with the underlying causal alleles. We introduced two complimentary approaches to better understand how the many individual variants are correlated with the underlying risk signals. We delineated each ancestral haplotype associated with disease, and we used linkage disequilibrium patterns to identify the variants that best detect independent risk signals carried by the haplotypes. To our knowledge, no prior investigation had reconstructed haplotypes comprehensively from all associated variants at a GWAS locus, nor explicitly used linkage disequilibrium patterns to identify sentinel variants.

We identified multiple overlapping 8q24 haplotypes with risk-altering effects, each distinguished by specific subsets of individual associated variants. These organize and provide context to previously identified 8q24 prostate cancer GWAS SNPs[18,20–23,27]. A given risk-altering haplotype can span adjacent LD blocks; haplotype A extends across the full 8q24 map. This could indicate the relatively recent introduction of a causal mutation, or the compound effect of causal variants on separate LD blocks that act jointly. An LD matrix reflects a great depth of mutational and recombinant population history, distinct from the history of a specific disease haplotype. Where such a haplotype is detectable within a collective study population, any given case may inherit some recombinant segment carrying a causal variant. An observation that we had not anticipated was the potential presence of more than one sentinel variant on a given identified risk-altering haplotype, each with statistically independent effects that indicate the presence of more than one causal variant. An individual's risk is a composite effect of inherited sentinels, some in close proximity that can co-segregate. At this locus, we observed that those in *cis* on a haplotype, and in *trans* as a diplotype interact multiplicatively.

Our LD-based RISSc algorithm for sentinel identification is tolerant of the number of variants under consideration and their correlations. It improves upon an alternative backwards stepwise regression approach, otherwise requiring prior reduction of the set via some additional forward selection method. The 8q24 sentinels include several known GWAS variants, two novel variants that better detect each of their corresponding risk signals than previously known variants, and one that is fully novel. We further compared these sentinels to a set of twelve sentinels identified in a recent publication that had used a forwards/backwards approach[26]. Sentinels rs1487240_G, rs77541621_A, and rs6983267_T were detected by both studies. We evaluated a merged model of the union set (17 total variants) among combined HPC cases and controls: 11 were significant under mutual adjustment (Supplementary Data File 3). Six of our eight sentinels remained significant in this merged model. Of the five variants that retained significance and that had originated from the other study, four had not been at genome-wide significance in our data, and so had not been considered by our algorithm. However rs5013678_C had been at genome-wide significance, but had not been selected; to explore, we relaxed the significance level for marking a given variant from $P \leq 0.01$ to $P \leq 0.05$ (see Methods), observing that rs5013678_C as well an additional variant (rs4871790_C) were then retained. This expanded our original set of eight to ten sentinels.

A facet that motivated our development of the RISSc algorithm is illustrated within the model evaluating all 17 variants of both sentinel sets. Any given pair of variants among them could be sufficiently correlated that neither retains significance under mutual adjustment, obscuring an independent risk-altering signal that is not captured by other variants. In this example, rs7832031_A (individual $P = 5 \times 10^{-13}$) and rs7812894_A (individual $P = 6 \times 10^{-13}$) have an $R^2 = 0.98$ and neither remains significant when modeled together, and yet they detect an independent risk signal. If either one of them is instead separately evaluated with the remaining 15 variants (Supplementary Data File 3), then the additional risk signal becomes visible. Analogously, two of three novel sentinels of Table 3 do not remain significant when jointly modeled with all known GWAS variants concurrently, and so do not appear in Table 2. However, explicit use of LD patterns discerns them as sentinels that better detect respective underlying risk signals than previously known variants with which they are correlated.

We identified eleven candidates for a causal mutation of near-Mendelian haplotype A, eight previously unknown. The functional effect of the causal mutation is likely to either alter a non-coding gene (the candidates in *PRNCR1* or *CASC19*) or a *cis* regulatory element (the identified candidates within enhancers). Risk-altering variants that act through a regulatory mechanism would be correlated with expression of an adjacent gene functioning in disease origin. Among the identified candidate mutations and sentinels, rs6983267 is an expression quantitative trait locus (eQTL) of *CASC8* in whole blood and of *POU5F1B* in colon[43]. We previously identified two 8q24 GWAS SNPs as eQTLs of *POU5F1B* in prostate, finding greater expression with the minor allele of sentinel rs6983267_T (marking protective haplotypes F and G), and with the minor allele of rs13252298_G (marking protective haplotype E)[29]. Comprehensive assessment of candidate mutation and sentinel variant function is needed. Where a variant has a particularly strong effect upon disease risk, a biological test of function could benefit from observation of a more binary assay outcome, in contrast to that expected of a variant of subtle risk effect.

The frequency and risk effect of 8q24 haplotype A is similar to *HOXB13* G84E[30]. Panel-based screening for hereditary cancer care encompasses *HOXB13*. Haplotype A could be considered for inclusion in such panels. The overall contribution of the 8q24 locus to prostate cancer risk includes sentinels of risk and protective effects, explaining ~9% of HPC heritability. In order to assess the potential clinical predictive utility of the 8q24 variants, we evaluated the area under the receiver operating characteristic curve (AUC) for a multivariable logistic regression model of 8q24 variants at genome-wide significance. The AUC of the combined HPC study populations was 0.661 (SE = 0.008). A model of the eight sentinel variants yielded an AUC = 0.635 (SE = 0.008); further adjustment for optimism of the model yielded a shrunk AUC of 0.625 ($P = 2 \times 10^{-65}$)[44,45]. 8q24 sentinels carry substantial predictive ability[46].

## Methods

**Study populations.** Cases of HPC pedigrees recruited for study include both PSA screen-detected and clinically-detected disease; potential lack of uniformity of screening standard of care across recruitment sites and over time presents an inherent limitation for any such observational study. Phenotypic heterogeneity is an existing feature of the disease, complicating its study.

**Nashville Familial Prostate Cancer Study (NFPCS).** The NFPCS is a case–control study. All subjects were recruited in the course of standard care at Vanderbilt University Medical Center and the adjacent Veteran's Administration Hospital between 2003 and 2009 under Institutional Review Board oversight and with written informed consent. This included men presenting for prostate cancer screening, with incident cases then treated for prostate cancer; other cases had been diagnosed at outside facilities and were referred for treatment. We recruited incident controls at the time of routine preventative screening for prostate cancer. The racial distribution of the study mirrored that of the study hospitals: 91% European and 9% African descent. A structured questionnaire was administered to

each subject at entry interview to elicit family cancer history (diagnoses, ages) and self-reported ancestry. Subjects were unselected for pedigree structure (e.g., number of brothers, male cousins). Case inclusion criteria required each prostate cancer case proband to have had one or more additional 1st or 2nd degree relatives with prostate cancer; 362 cases met stricter criteria for HPC. Each HPC case proband had two or more additional 1st or 2nd degree relatives with prostate cancer. HPC case probands included 331 of European ancestry[4,12]. A total of 340 case probands of European ancestry instead were diagnosed under the US national mean of 66 years, and were each from a pedigree with only one additional affected male relative. We refer to these men as having familial prostate cancer (FPC) to distinguish them from HPC cases. The European ancestry HPC case probands were employed for all analyses, while the FPC case probands served as an auxiliary set for comparison. Controls were required to have: a negative personal and family history of prostate cancer, no known abnormal digital rectal examination, no prior prostate biopsy, a screening prostate specific antigen (PSA) level below 4 ng ml$^{-1}$ (93% were below 3 ng ml$^{-1}$), and all prior known PSA levels also below this level. Controls were unselected for pedigree structure (e.g., number of at-risk relatives or their ages, though the mean number of brothers of cases was 1.7 and of controls was 1.8). Among European ancestry subjects evaluated in the study, the age profile of the 823 controls was older than that of cases (Table 1), which is conservative (genotype and phenotype would be identical if a given control had been recruited at the same age as a younger case). Adjusting for age had no meaningful effect on our results. In parallel with case–control recruitment, we separately ascertained multiple affected men within each of 32 European ancestry HPC pedigrees (only the proband for each is tallied in Table 1). For these pedigrees, we genotyped a mean of three affected men per family to enable secondary assessment of segregation. The NFPCS has previously demonstrated ability to detect both rare, large-effect and common, small-effect risk variants[16,20,21,29,41,47].

Abstracted clinical data included age at diagnosis or screen, PSA level at diagnosis or screen, clinical TNM, pathologic TNM, left and right lobe Gleason scores and sum, extracapsular extension, seminal vesical invasion, and margin status. Radical prostatectomy was the treatment modality for 97% of case subjects, providing definitive pathologic grade and stage; 21% had extra-prostatic disease (pT3, pT4, N1, or M1). Germline DNA was prepared from whole blood for each subject using the Puregene DNA Purification System Standard Protocol (Qiagen) on an Autopure LS robot, and quantified by PicoGreen (Invitrogen). NFPCS samples were genotyped using the Illumina Multi-Ethnic Genotyping Array, including blinded duplicate study samples and HapMap trios for quality control. Array data was processed using GenomeStudio analysis software with de novo clustering and quality control using the pipeline of the Vanderbilt genomics core and biobank[48]. Orthogonal genotype assays were completed by commercial TaqMan (Life Technologies)[49] assay or by custom assay using single-nucleotide primer extension with detection by fluorescence polarization[50].

**International consortium for Prostate Cancer Genetics Study.** Data of the International Consortium for Prostate Cancer Genetics (ICPCG) GWAS of Familial Prostate Cancer was obtained from dbGaP, accession phs000733.v1.p1 (Principal Investigator Lisa Cannon-Albright, PhD). Case and control selection criteria are previously published and detailed in dbGaP meta-data[22]. The data set encompasses 2505 analyzed independent HPC cases aggregated from 12 separate studies conducted at the following sites: Cancer Council Victoria/Australia (Drs. Graham Giles and Liesel Fitzgerald), the Center for Research on Prostatic Diseases/France (Drs. Geraldine Cancel-Tassin and Olivier Cussenot), the Fred Hutchinson Cancer Research Center (Drs. Elaine Ostrander and Janet Stanford), the Institute of Cancer Research/UK (Drs. Rosalind Eeles and Zsofia Kote-Jarai), Johns Hopkins University (Dr. William Isaacs), Louisiana State University (Dr. Diptasri Mandal), the Mayo Clinic (Drs. Daniel Schaid and Stephen Thibodeau), Northwestern University (Dr. William Catalona), Tampere University/Finland (Dr. Johanna Schleutker), the University of Michigan (Drs. Kathleen Cooney and Ethan Lange), and the University of Ulm/Germany (Dr. Christiane Maier), and the University of Utah (Drs. Lisa Cannon-Albright and Craig Teerlink). Each case was selected from a previously ascertained pedigree with ≥3 affected men.

Among the sampled cases of a given independent pedigree, one with the most aggressive disease or earliest age of diagnosis was selected for study. Age at diagnosis and categorical severity (aggressive, moderate, and insignificant) accompanies each case. Categorical severity definitions are given in dbGaP meta-data and are summarized below. Nine sites contributed 1383 analyzed unrelated male controls without a cancer diagnosis. Ages of ICPCG controls were of similar frequency distribution to cases[22], though not individually recorded in the data set. Prior ICPCG analyses did not adjust for age[22]. All subjects were of self-reported European ancestry. Genotype data was generated by the Center for Inherited Disease Research (CIDR) using the Omni5Exome array. The approach described above was used to process array image files with de novo clustering and subsequent quality control.

**Imputation.** The University of Michigan Imputation Server pipeline was employed for genotype imputation. Array-generated genotypes were used as the basis for imputation against reference whole genome sequence of subjects of European descent from the Haplotype Reference Consortium r1.1 2016. Phasing employed Eagle v2.3 with imputation using Minimac 3. Only informative bi-allelic SNPs were used as the basis for imputation, with a required minimum genotype completion rate of 98% and subject completion rate of 98%. The most probable genotypes for imputed variants with an estimated accuracy of $R^2 \geq 0.8$ were analyzed for association with prostate cancer among ICPCG subjects.

**Genetic ancestry.** All evaluated subjects of the NFPCS and ICPCG subjects were of self-reported European ancestry; we confirmed subject genetic independence as well as genetic ancestry. We conducted a principal components analysis to enable statistical adjustment for genetic ancestry, given potential for subtle genetic ancestry differences across the distinct global recruitment sites. We used FlashPCA2.0 and pruned, post-imputation genome-wide genotype data (40,964 variants of MAF ≥ 0.01, Hardy-Weinberg equilibrium $P \leq 5 \times 10^{-5}$, genotype missingness ≤ 0.001, pairwise $R^2 \leq 0.02$) to calculate principal components[51].

**Statistical analyses.** Genetic association with disease was tested by analyzing individual variants to first identify associations that replicated across the two independent study data sets, with subsequent further assessment using haplotype-based tests. Unconditional multiplicative logistic regression models were employed. These models were additive on the logit scale (additive genetic models). Significance was assessed using Wald tests. When genotype perfectly predicted case–control status, we instead use Choi's likelihood ratio chi-squared test of association[52]. An association between genotype and disease was considered nominally significant with a two-sided $P \leq 0.05$. A total of 765 nominally significant variants within ICPCG data were subsequently evaluated in NFPCS data. Benjamini and Hochberg false discovery rates in the NFPCS were derived for these NFPCS tests[53]. A Bonferroni-corrected significance level of $P < 7 \times 10^{-5}$ corresponds to 765 (non-independent) tests, while $P < 2 \times 10^{-4}$ corresponds to tests of their 242 linkage disequilibrium bins of $R^2 \geq 0.8$. In order to assess the effect of age as a potential confounder in the NFPCS, we additionally evaluated models adjusted for age.

Data of the ICPCG and NFPCS were combined to further assess significance and effect size of replicating variants. Associations were considered to be genome-wide significant by the convention of $P \leq 5 \times 10^{-8}$. We further evaluated these variants in logistic regression models adjusted for the first four principal components of genetic ancestry. Adjusting for principal components had no meaningful effect on association results.

The proportion of trait variance explained by a set of genetic variants ($h^2_{\text{SNP}}$) was tested by the restricted maximum likelihood approach implemented in GCTA v1.90.2[33]. Heritability estimates were assessed in study HPC cases and controls. Estimates were transformed to the liability scale using the NCI SEER estimated lifetime prostate cancer risk of 11.6%[54–56]. Results were not substantively altered by higher or lower prevalence estimates, or by adjustment for the first four principal components of genetic ancestry. For example, even assuming a lifetime risk of 0.80 for men from HPC pedigrees[57], the heritability explained by the 433 concordantly significant variants among combined ICPCG and NFPCS HPC subjects was 10.8% (SE = 0.027); under further adjustment for genetic ancestry, it was 10.6% (SE 0.028). Limitations by this approach are discussed in the published literature[54,58–62]. Factors that could influence accuracy include study power, trait misspecification, genotype error, or missingness, poor representation of causal alleles by LD, cryptic relatedness, or a disease architecture of causal alleles departing from an additive model. Bayesian variable selection regression (BVSR) implemented in piMass v0.90 was used as an alternative method to calculate variant posterior inclusion probabilities as well as the proportion of trait variance explained[34]. For BVSR we obtained 10 million Markov Chain Monte Carlo samples from the joint posterior probability distribution of model parameters (recording values every 400 iterations), discarded one million samples as burn-in, and set bounds on heritability between 0.01 and 0.58.

The set of 433 variants that were nominally significant in both the ICPCG and NFPCS were further investigated in haplotype-based analyses. Subject diplotypes were imputed using Phase v2.1[63,64] with a 0.9 probability threshold for subsequent tests of association with disease. We performed a sliding-window haplotype analysis, with window width varying from two to 20 adjacent variants[29,31]. Windows were incrementally moved along the map, and haplotypes of a given window were tested for disease association under additive genetic models. Only window haplotypes that were concordantly significant in each of the two study populations separately ($P \leq 0.05$ with consistent direction of effect) were retained. Those uniquely aligning in overlapping windows formed a summary model of a given identified risk-altering haplotype. Observations were confirmed using Beagle v3.1.0[65] as an independent phasing algorithm. A secondary analysis was also conducted separately for each identified risk-altering haplotype; we constrained the variants tested to only the subset with minor alleles marking the haplotype of interest (each minor allele with a direction of effect consistent with that of the haplotype carrying it).

We tested the 433 variants that were nominally concordantly associated with disease in the two study populations as potential modifiers of age at diagnosis. A Wilcoxon rank-sum test was used to test association between genotype and age of diagnosis among cases of the combined studies. We also investigated these variants as potential modifiers of prostate cancer severity. Cases were categorized into severity groups mirroring ICPCG criteria (Aggressive: extra-prostatic stage at diagnosis (≥T3, N1, or M1), or Gleason ≥ 8 (poorly differentiated), or PSA ≥ 20 ng ml$^{-1}$ at diagnosis,

or lethal prostate cancer; Insignificant: stage T1 or in only one lobe (T2a) if prostatectomy done, and no evidence of extra-prostatic disease, and Gleason ≤ 6 (not moderately or poorly differentiated), and PSA ≤ 4 ng ml$^{-1}$ at diagnosis, and if deceased did not die of prostate cancer; Moderate: cases not meeting either aggressive or insignificant criteria).

Cases of each pair of these severity groups were evaluated in dichotomized comparisons (aggressive vs. insignificant, moderate vs. insignificant, and aggressive vs. moderate) by logistic regression models both with and without adjustment for the first four principal components of genetic ancestry. Reported results are those without adjustment (results with adjustment were more significant).

**Selection of sentinel SNPs**. We designed the following recursively identified sentinel scoring (RISSc) algorithm to identify sentinel SNPs among the 183 that individually achieved genome-wide significance in the combined studies. This algorithm selects SNPs that are mutually significant in a multivariable model, and which have low pair-wise $R^2$ values. These are sentinel SNPs, optimally detecting the independent risk-altering association signals of the starting SNP set. In what follows, all regressions are logistic and use multiplicative (additive genetic) models; *case* is an indicator variable that identifies HPC cases and controls. The algorithm identifies bins of SNPs that are correlated with each other with diminishing $R^2$ thresholds. "Selected" means kept for possible consideration in the final sentinel model. A selected SNP is "marked" if its association with disease is sufficient to keep it from being deleted in the next step. Not all marked SNPs will make it into the final model. Once a SNP is deleted, however, it is permanently excluded from further consideration for inclusion in the final model.

Step 1: Identify bins of SNPs that are perfectly correlated with each other ($R^2 = 1$). Select one SNP from each bin and delete all other SNPs in each bin from further consideration. Bins of size 1 are allowed. Regress *case* against all selected SNPs in a multivariable logistic regression model. If this regression converges then mark all selected SNPs of $P \leq 0.01$ for further consideration and designate those of $P > 0.01$ as unmarked. If the regression does not converge, then all selected SNPs are unmarked but remain as candidates for further evaluation. Set $R^2 = 0.975$. Proceed to Step 2 with the selected SNPs, each categorized as either marked or unmarked.

Step i: $i = 2$–$40$ Identify bins of selected SNPs from Step $i-1$ whose squared correlation coefficient is $\geq R^2$. For each bin:

a. Identify the SNP with the greatest association with disease using simple logistic regression. This SNP is denoted "best in bin."
b. Regress *case* against all of the SNPs in the bin. The best in bin SNP plus any SNP in the multivariable regression for this bin that has $P \leq 0.01$ are selected together with all SNPs that were marked in Step $i-1$. Delete all SNPs in the bin that have not been selected from further consideration. After the selections and deletions from each bin have been made, regress *case* against all of these remaining SNPs in a multivariable logistic regression model. If this regression converges, then mark all SNPs of $P \leq 0.01$ while designating those of $P > 0.01$ as unmarked. Any SNP that was previously marked will become unmarked if it no longer meets this $P$-value threshold. If the model instead fails to converge, then retain the modeled SNPs but designate them as unmarked unless they were marked at the previous step. Subtract 0.025 from $R^2$ and increment $i$ by 1. If $i \leq 40$ loop to repeat Step $i$.

The final sentinel SNPs identified by this algorithm are those that were marked in Step 40. In application to the 183 genome-wide significant variant set, failure of a multivariable model to converge was only observed at Step 1 (SNPs representing bins of $R^2 = 1$).

**Optimism of the sentinel SNP model**. We regress *case* against the SNPs identified in the preceding algorithm to calculate the area under the receiver operator characteristic curve (AUC) associated with the linear predictor from this model. We ran 1000 bootstraps of this algorithm to derive a bootstrapped aggregated (bagged) estimate of the optimism of our sentinel SNP model[44,45]. For each bootstrapped sample we re-ran the sentinel algorithm to obtain the AUC estimate from this sample (AUCboot). We then applied the parameter estimates from this model to the real data to obtain an AUC using the bootstrapped parameter estimates (AUCreal). The optimism estimate from this bootstrapped sample is AUCboot – AUCreal. The bagged estimate of the optimism of our algorithm is the average value of optimism estimate from the 1000 bootstrapped samples. The shrunken estimate of the AUC (that reported within the manuscript) is the AUC from the real data minus the bagged optimism estimate. This shrunken estimate adjusts for overfitting of the sentinel model that may occur due to the number of variants considered by the RISSc algorithm.

**Whole genome sequencing**. Sequencing was performed by the University of Washington Center for Mendelian Genomics with 150 bp paired-end reads. DNA was sheared targeting 380 bp inserts for library construction, with AMPure XP bead cleanup, sample prep using the Roche KAPA HyperPrep kit and a two-sided AMPcure cleanup to further select fragment size, with subsequent end-repair, A-tailing, ligation, and then excess adapter removal by a

final AMPure cleanup. Libraries were validated using a Biorad CFX384 Real-Time System and KAPA Library Quantification Kit. Barcoded libraries were pooled for clustering on an Illumina cBot. Sequencing by synthesis was done on an Illumina NovaSeq.

The processing pipeline consisted of the following elements: (1) base calls generated in real-time on the NovaSeq6000 (RTA 3.1.5); (2) demultiplexed, unaligned BAM files produced by Picard ExtractIlluminaBarcodes and IlluminaBasecallsToSam; and (3) BAM files aligned to a human reference (hg19hs37d5) using BWA[66] (Burrows-Wheeler Aligner; v0.7.10). All aligned read data were subject to the following steps: (1) "duplicate removal" was performed, (i.e., the removal of reads with duplicate start positions; Picard MarkDuplicates; v1.111); (2) indel realignment was performed (GATK IndelRealigner; v3.2-2) resulting in improved base placement and lower false variant calls; and (3) base qualities were recalibrated (GATK[67] BaseRecalibrator; v3.2-2).

Variant detection was performed using the HaplotypeCaller tool from GATK (3.7). Variant data for each sample were formatted (variant call format [VCF]) as "raw" calls that contain individual genotype data for one or multiple samples and flagged using the filtration walker (GATK) to mark sites that are of lower quality/ false positives [e.g., low quality scores (Q50), allelic imbalance (ABHet 0.75), long homopolymer runs (HRun > 3) and/or low quality by depth (QD < 5)].

Data quality control included assessments of: mean coverage, fraction of genome covered greater than 10×, duplicate rate, mean insert size, contamination ratio, mean Q20 base coverage, transition/transversion ratio, fingerprint concordance > 99%, sample homozygosity and heterozygosity, and sample contamination validation. We obtained 92% genome coverage with a mean of 48.6-fold (1.03B reads per sample). The mean read length was 151 bp with a mean paired-end read span of 394 bp. Phase was inferred by allele sharing among the sequenced haplotype A carriers, confirmed by paired-end reads spanning adjacent variants.

The SeattleSeq Annotation pipeline was used for annotation, returning: rsID (or novel), gene name and accession number, predicted functional effect (e.g., splice-site, nonsynonymous, missense, etc.), protein positions and amino-acid changes, PolyPhen predictions, Grantham score, CADD score, PhastCons and GERP conservation scores, cpg islands, transcription factor binding sites, dbSNP allele frequencies, and known clinical associations. Read data of all variants within the 648 kb 8q24 interval were manually reviewed using CLC Genomics Workbench v11.

**Reporting summary**. Further information on research design is available in the Nature Research Reporting Summary linked to this article.

## Data availability

The source data underlying Fig. 2 as well as Tables 2, 3 are provided within the Source Data file (tab 1). Underlying source data of the NFPCS in Fig. 3 is provided in the Source Data file (tab 2), while data for the ICPCG is accessioned from the dbGaP link below. Supplementary Data File 2 provides variant calls of sequenced samples, with detail for mutation candidates. Data pertaining to the International Consortium for Prostate Cancer Genetics (ICPCG) GWAS of Familial Prostate Cancer was obtained from dbGaP under the accession code phs000733.v1.p1. Other datasets referenced during the study are available from respective websites: gnomAD [https://gnomad.broadinstitute.org], GWAS Catalog [https://www.ebi.ac.uk/gwas/search], The International Genome Sample Resource (1000 Genomes) [https://www.internationalgenome.org/data#download], SeattleSeq Annotation [https://gvsbatch.gs.washington.edu/SeattleSeqAnnotation138/index.jsp], TOPMed [https://bravo.sph.umich.edu/freeze5/hg38/], and the University of Michigan Imputation Server (and encompassed Haplotype Reference Consortium r1.1 2016 reference) [https://imputationserver.sph.umich.edu/index.html#!pages/home]. Other data supporting the findings of this study are available within the article and its supplementary information files and from the corresponding author upon reasonable request. A reporting summary for this article is available as a Supplementary Information file.

## Code availability

RISSc software for sentinel variant identification is open source and available from the IDEAS/RePEc Statistical Software Components archive [https://ideas.repec.org/s/boc/bocode.html], which is the primary repository of contributed STATA programs. It is also available at http://biostat.mc.vanderbilt.edu/.

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

## Acknowledgements

We wish to thank investigators of the ICPCG for sharing the data set of dbGaP accession phs000733.v1.p1. ICPCG investigators are: Drs. Joan Bailey-Wilson, Geraldine Cancel-Tassin, Lisa Cannon-Albright, John Carpten, William Catalona, Kathleen Cooney, Olivier Cussenot, Rosalind Eeles, Liesel Fitzgerald, Graham Giles, Chih-Lin Hsieh, William Isaacs, Zsofia Kote-Jarai, Ethan Lange, Diptasri Mandal, Christiane Maier, Elaine Ostrander, Isaac Powell, Daniel Schaid, Johanna Schleutker, Janet Stanford, Craig Teerlink, Stephen Thibodeau, Alice Whittemore, Fredrik Wiklund, and Jianfeng Xu. We also wish to thank the participating patients. This study was supported by awards from the Veteran's Administration, the V Foundation, and National Institutes of Health grants UL1 TR000445 and P30 CA068485. Sequencing was done by the University of Washington Center for Mendelian Genomics (UW-CMG) and was funded by NHGRI and NHLBI grants UM1 HG006493 and U24 HG008956 and by the Office of the Director, NIH under Award Number S10OD021553. The content is solely the responsibility of the authors and does not necessarily represent the official views of the funding agencies.

## Author contributions

J.R.S. conducted the NFPCS with contributing subject referral by S.S.C., M.S.C., and J.A.S. J.P.B. and J.R.S. performed genotyping and quality control of data. M.J.B., E.E.B, and the UW-CMG performed whole genome sequencing. J.P.B., J.R.S., and UW-CMG curated sequence data. W.D.D., W.D.P., and J.R.S. performed statistical analyses. J.R.S. designed the study. All authors contributed to the manuscript.

## Competing interests

The authors declare no competing interests.

## Additional information

## University of Washington Center for Mendelian Genomics

Elizabeth Blue [6], Michael Bamshad[7], Jessica Chong [8] & Deborah Nickerson[9]

[8]Department of Pediatrics, University of Washington, 1959 Pacific Street, HSB I607F, Seattle, WA 98195, USA. [9]Department of Genome Sciences, University of Washington, 3720 15th Avenue NE, Foege S-213B, Seattle, WA 98195, USA.

