## [Peer Review File · Nature Communications]

Reviewers' comments:

Reviewer #1 (Remarks to the Author):

This is an interesting and well written manuscript.

I did not see any information in the paper about the distribution of numbers of relatives with prostate cancer or the definition of 'familial'. Were the genetic factors more prominent in the more familial cases? Also, its not clear how analysis of familial cases can identify protective factors.

I am wondering if you were to analyze nonfamilial cases if there would be more protective haplotypes in that population. As a referrent it may be helpful to analyze a set of unselected controls from similar populations but sequenced on the same platforms.

Along those lines one of the more prominent findings was an STR. STRs can be hard to sequence on next generation sequencing and more detail about the protocol for genotype calling for the STR would make the results more easy to reproduce.

The approaches to analysis generally seem well developed and well conducted. It was a little confusing that the AUCreal in the methods section is referred to by a different name in the results (aligning the names would be helpful).

Reviewer #2 (Remarks to the Author):

Review for Dupont et al.,

The authors explored the 8q24 locus, which has been tied to a number of cancers, most particularly hereditary forms of prostate cancer. The study identified multiple variants, independently and multiplicatively associated with both risk and protective effects.

Th is an excellent, if overly complex paper whose real value of this paper is in the definition of multiple risk haplotypes at 8q24, rather than the identification of novel alleles, as implied early in the paper. What is really needed is clarity. The abstract should, for instance, clearly state how many haplotypes are found (7), how many SNPs define each, and what the size of each is and the overlap. Ideally this would also include a population-based summary and phenotype summary. Much of that is in the paper, on page 9, but clarify and organization is needed. It is hard to distill.

The investigators compared 2505 independent hereditary prostate cancer cases (figure 1) from the ICPCG study to 1383 controls. This is confusing because the abstract says 2836 and 2206 controls. Please clarify.

The study identified a set of rare variants and then imputed from reference WGS of European ancestry. They identified 20 variants that met genome wide significance (<0.05). Figure 1 is confusing because they refer in the second box to the NFPCS study of 331 hereditary cases and 823 controls and identify 443 variants of which 55 are novel for a total of 183, of which six are novel. Analysis of the seven multiplicative sentinel variants reveals two novel variants. The 183 are predicted to explain 8-15% of PCA risk. It is unclear what the 765 is for in box two in column one of figure one.

The authors assessed the minor allele of all 433 risk variants to their corresponding haplotypes and showed that there were four risk haplotypes and three protective haplotypes. One risk haplotype was associated with early age at diagnosis while a projective one was associated with later age at diagnosis. They also sequenced carriers of a "near-Mendelian" risk haplotype and identified three previously known candidates for causal mutations. In a second approach they identified independent putative risk among the many identified at genome wide significance using

a new LD based algorithm. They identified eight sentinel variants, three of which are novel and carry risk prediction ability at a single locus.

When considering the 13 of 765 variants they authors say that 13 are genome wide significant. The real issue is not how many novel variants were found, but how many define new haplotypes. The presence of one new variant on a haplotype defined by several known variants is of much less interest unless it expands the haplotype, which is not discussed. The analysis of genetic variants residing on seven risk-altering haplotypes is the heart of the paper but is lost in the dribbling out of information regarding those haplotypes and novelty of individual markers.

The Nashville study are men that lack a personal or family history of PC. What does that mean? Do you have actual recent PSA data on these men and family history interviews? Did you remove men without sufficient first-degree relatives to have a family history of disease?

The addition of the 343 additional cases of early onset really confused the study. Each of these men had a weak family history of disease. Did any of the cohort selections include selection of men with aggressive disease? Did any include selection for early onset disease (outside of the above mentioned).

It would be useful to break up the OR for the 200 with an OR between 1 and 2. A ratio of 1.89 means much more than 1.1.

Analysis in the combined populations is worrisome. The studies are very different. Please discuss limitations of doing this.

Reference 30 which discusses the odds ratios for BRCA1/2 is for breast cancer study not prostate as implied. Clarify the sentence.

The authors used cases from the ICPCG study. ICPCG is somewhat flawed in that they collected families whose evidence for "hereditary" falls short. Little consideration was given to degree of aggressiveness in the original collection. This can be dealt with using a table in the supplement that clarifies how cases were picked and defining groups of cases sharing characteristics. Other than early onset, do the haplotypes associate with other phenotypes? Response to therapy, outcome, aggressive disease?

Figure 3 is excellent and helpful.

Comparison on numbers of men in families carrying high risk haplotypes to HOX13 G84E are very useful and correct.

Is the 0.6 year later age at onset significant? Given that men were diagnosed during the PSA onset era—thus some were diagnosed by RDE with large tumors and others very early in their disease by rising PSA (which is never discussed), how significant can 0.6 years be?

Sentinel variant analysis is interesting and applicable to many studies. It is a strength of the paper.

For the sequencing of seven cases with haplotype A, please provide more information at the onset of the section on the families and cases themselves (age at diagnosis, degree of family history, etc.), rather than sharing it piecemeal. Put a table in the supplement. This analysis suffers from a lack of discussion regarding the likely functional impact of each variant identified as being a candidate for causal. Have the authors analyzed likely functionality using REVEL or other similar programs? Please include analysis of the four that are not discussed.

Some references are out of date. The claim that analysis of hundreds of pedigrees meeting the

criteria for HPC have been examined with little success since the 1993 Carter paper. Considerably more work has been done, some with success, and should be cited. Also the citation of one paper showing a GWAS with a significant GWAS associated at 8q24 does not do justice to the literature, as several have been published and the paper cited focuses on just one variant.

In the first paragraph of the discussion are the authors saying that the NFPCS study was the first to conduct a study focused on family history or hereditary cases? That is incorrect. The Hopkins study identifying the HPC1 locus was first. If the authors are saying something else, please clarify.

Identification of private variants in individual families are unlikely to be found by linkage, as claimed on page 14. Individual families are way too small to achieve significance and the degree of locus heterogeneity makes it impossible (see example of HOXB13).

The fact that some haplotypes are inclusive, or span others is important and should be better highlighted on page 15. How much does this affect the risk analysis?

Overall the paper does a poor job of putting these findings into the context of the published large prostate cancer GWAS, which generally are not family based, but do include family history as a variable. While those studies did not do haplotype analysis and lack the originality and compelling results here, to not even mention them and their identification of hundreds of variants is an omission from the discussion.

Having said all that, this is a really clever and much needed analysis that conveys new information to the field. What is needed is brevity and clarity.

October 22, 2019

Anonymous Reviewers

Dear Reviewers,

We would like to thank you for positive and thoughtful review of our manuscript titled "Novel 8q24 genetic variation and comprehensive haplotypes altering familial risk of prostate cancer" (NCOMMS-19-16639-T).

Our responses to each of your comments are given below. The revised text has been edited to incorporate numerous facets of these comments, designated by page and line number where appropriate.

Reviewer 1.

Definition of "familial." "Hereditary" prostate cancer (HPC) has an established definition that has been employed in global studies since its introduction^{1,2}. Page 3 line 51 of the Introduction and page 12 line 479 of Methods define HPC as a case from a family with three or more affected first- or second-degree relatives. Manuscript tables and figures analyze cases meeting this criterion. Supplementary Figure 1, and the second tab of Supplementary Table 2 further evaluate an additional group of cases. The additional case group instead meets the following criteria: 1) was diagnosed under the national mean age of 66 years, and 2) was from a family with a single additional affected first- or second-degree relative. These are defined as "familial" cases (FPC) on page 4 line 121 (Results) and page 12 line 481 (Methods). The count distributions are also provided.

Were the genetic factors more prominent in the more familial cases? In general, yes. As an example, sentinel rs77541621 of intermediate-risk haplotype B has an allele frequency of 7.27% in NFPCS HPC cases, 6.16% in NFPCS FPC cases, and 1.95% in NFPCS controls. Although outside the scope of this manuscript, the frequencies of the variant in the PLCO GWAS³ were: 3.97% in cases unselected for family history, and 2.09% in controls. Strength of family history is correlated with degree of risk variant enrichment, and the enrichment skew is greater for rarer, larger-effect variants.

It's not clear how analysis of familial cases can identify protective factors. Men with a strong family history of disease are expected to have an excess of genetic risk factors, and a lack of protective ones. These differences are relative to controls. Protective factors are those in measured excess among controls relative to cases (page 5 line 157).

At a hypothetical GWAS SNP, the minor allele may have an $OR = 0.5$ and $P = 5E-8$, and the major allele will then necessarily have an $OR = 2$ and $P = 5E-8$. This obligate yin-yang of a bi-allelic SNP

effect can be disentangled by haplotype analysis. Which allele is specific to a haplotype of significant effect, and what is the direction of that effect? Many GWAS SNPs have a minor allele in excess in controls relative to cases, though the corresponding publications have often been presented as major allele conferring risk (rs6983267 is one example).

I am wondering if you were to analyze nonfamilial cases if there would be more protective haplotypes in that population.

This study evaluated only familial cases. We would expect protective haplotypes (and their sentinels) to be in less prominent excess among controls relative to non-familial cases, though this topic is beyond the scope of the manuscript. Using the external PLCO trial for comparison, protective sentinel rs12678349 had an unselected PLCO case frequency of 8.26% and a PLCO control frequency of 9.82%. In our study, the HPC case frequency was 6.95% and control frequency was 10.38%.

As a referent It may be helpful to analyze a set of unselected controls from similar populations.

Our study did not employ cosmopolitan controls, among whom disease status would be unknown. The gnomAD reference (<https://gnomad.broadinstitute.org>) is a resource providing general population frequencies. Each of the three protective sentinel alleles of Table 3 are at modestly greater frequency among screened study controls than reported among gnomAD non-Finnish Europeans. Absence of disease in a subject group is relatively less selective as a criterion (for controls) than presence of disease as well as presence of a family history for the disease (for cases). In general, screened controls are more subtly different from the population from which they are drawn, relative to the difference between familial cases and the general population.

STRs can be hard to sequence on next generation sequencing and more detail about the protocol for genotype calling for STRs would make the results more easy to reproduce.

Alleles were called using the pipeline of the Center for Mendelian Genomics at the University of Washington, described in Methods. The Center performs sequencing for the Undiagnosed Disease Network. We have expanded description detail within this methods section (page 17, beginning line 688). STR alleles within sequence data of Haplotype A carriers (Table 4, Supplementary Table 2) were unambiguous on direct review of reads in CLC Genomics Workbench. Alleles carried by haplotype A were consistent across the sequenced cases. We should note that all sequence variants were manually reviewed, and that the allele presentation of Supplementary Table 2 was extensively curated.

It was a little confusing that the AUC_{real} in the methods section is referred to by a different name in the results (aligning the names would be helpful).

We have revised the pertinent sentence of Methods for clarity (page 16, line 673): "The shrunken estimate of the AUC (that reported within the manuscript) is the AUC from the real data minus the bagged optimism estimate. This shrunken estimate adjusts for overfitting of the sentinel model that may occur due to the number of variants considered by the RISSc algorithm."

Reviewer 2.

This an excellent, if overly complex paper whose real value is in the definition of multiple risk haplotypes at 8q24, rather than the identification of novel alleles, as implied early in the paper. What is really needed is clarity. The abstract should clearly state how many haplotypes are found (7), how many SNPs define each, and what the size of each is and the overlap.

The manuscript deconstructs a complex locus. Many readers expect a typical presentation that is built around novel individual alleles, motivating that aspect. However, we concur that haplotype organization and sentinels are the manuscript's greater contribution.

A 150-word abstract is very limiting. We revised the abstract to state: "The variants comprehensively distinguish seven independent risk-altering haplotypes overlapping the 648 kb locus (three protective

and four risk (peak odds ratios: 1.5, 4, 5, and 22))." Abstract brevity does not accommodate further detail without omission of other important content.

The investigators compared 2505 independent hereditary prostate cancer cases (Figure 1) from the ICPCG study to 1383 controls. This is confusing because the abstract says 2836 and 2206 controls. Please clarify. Ideally [the abstract] would also include a population-based summary and phenotype summary.

We modified the abstract: "We investigate its contribution to hereditary prostate cancer (HPC) in independent study populations of the Nashville Familial Prostate Cancer Study and International Consortium for Prostate Cancer Genetics (combined: 2,836 HPC cases, 2,206 controls of European ancestry)." Figure 1 (and Table 1) provides the separate subject counts of the NFPCS and of the ICPCG.

Figure 1 is confusing because it refers in the second box to the NFPCS study of 331 hereditary cases and 823 controls and identify 433 variants of which 55 are novel for a total of 183, of which six are novel. It is unclear what the 765 is for in box two in column one of figure one.

We have revised Figure 1 for clarity:

The first box conveys that a set of 765 variants at 8q24 were observed that were each nominally significantly associated with prostate cancer ($P < 0.05$) in ICPCG subjects.

The second box summarizes that the 765 variants were carried forward from box 1 for evaluation of replication in the NFPCS, yielding 433 that were also significant in the NFPCS.

The third box summarizes that the 433 replicating variants were then evaluated in subjects of both studies combined, observing that 183 met genome-wide significance.

The presence of one new variant on a haplotype defined by several known variants is of much less interest unless it expands the haplotype, which is not discussed. The analysis of genetic variants residing on seven risk-altering haplotypes is the heart of the paper but is lost in the dribbling out of information regarding those haplotypes and novelty of individual markers.

The manuscript reveals over-arching organization as haplotypes, and further distills the relatively large number of individual risk-altering variants (known and novel) to the set of sentinels that best detect the underlying risk-altering signals carried by each haplotype. The presented relationship between both known and novel sentinels and the larger haplotypes is a valuable focal point of the manuscript. We have included a diversity of presentation topics that will each be of interest to various readers. Individual marker novelty was, for example, explicitly sought by readers of the draft manuscript.

The Nashville study are men that lack a personal or family history of PC. What does that mean? Do you have actual recent PSA data on these men and family history interviews? Did you remove men without sufficient first-degree relatives to have a family history of disease?

A structured questionnaire was administered in person by an experienced genetic counselor to elicit the family history of all subjects, including controls. Controls had a negative family history of prostate cancer among first- and second-degree relatives. Subjects were unselected for pedigree structure (page 12 line 476). The mean number of brothers of cases is 1.6 and of controls is 1.7. Controls were screened for prostate cancer in the course of routine medical care; they were recruited at the prostate cancer screening visit which included PSA testing. Controls were additionally required to have no prior known abnormal DRE, elevated PSA, or any prior prostate biopsy. These aspects of NFPCS subject selection are described beginning on page 12 line 465.

The age profile of controls was older than that of cases (Table 1), which is conservative (genotype and phenotype would be identical if a given control had been recruited at the same age as a younger case). Adjusting for age had no meaningful effect on results.

The addition of the 343 additional cases of early onset confused the study. Each of these men had a weak family history of disease. Did any include selection for early onset disease (outside of the above mentioned).

We reserved the separate group of FPC cases for secondary comparative analyses. These analyses are explicitly described, and distinguished from the primary analyses of HPC cases. As noted above, genetic risk factors are enriched even in the FPC group.

HPC cases of the NFPCS were not selected for age of diagnosis. A subset of ICPCG HPC cases are reported to have been selected based upon age of diagnosis (Methods page 13 line 533). However, the age distribution of NFPCS and ICPCG HPC cases are comparable (Table 1).

Did any of the cohort selections include selection of men with aggressive disease? HPC is distinguished by an early age of diagnosis, but not by pathological features. NFPCS cases were not selected based upon pathological severity (97% of cases were treated by radical prostatectomy, providing definitive pathology data for the study). A subset of ICPCG cases are reported to have been selected for study as the most aggressive case of the represented pedigree (Methods page 13 line 533). ICPCG case data includes only categorical severity (page 13 line 534). Categorical severity of combined NFPCS and ICPCG cases was not associated with 8q24 variants (page 9 line 359).

It would be useful to break up the OR for the 200 with an OR between 1 and 2. A ratio of 1.89 means much more than 1.1.

The revised corresponding sentence of page 5 line 132 now states: "Odds ratios (ORs) of minor alleles may be partitioned by effect size: 8 of OR > 4; 68 of OR between 2 and 4; 43 of OR between 1.5 and 2; 157 of OR between 1 and 1.5; and 157 of OR < 1 (**Figure 2 and Supplementary Table 1**)."

Analysis in combined populations is worrisome. The studies are very different. Please discuss limitations of doing this.

We have expanded discussion of potential advantages and limitations to include (page 9 line 351): "Potential genetic heterogeneity across subjects recruited from different global locations represents another potential limitation. Adjustment of analyses for genetic ancestry would guard against confounding due to population structure, but a disease variant specific to one of the component study populations might have reduced power for detection or fail to replicate. Potential phenotypic heterogeneity across different recruitment sites is another limitation for any such study. Family history criteria for HPC cases were uniform across sites; however, subsets of ICPCG cases were also further selected based upon disease aggressiveness or age of diagnosis. Two 8q24 haplotypes were associated with age of diagnosis, but we did not observe association with aggressiveness (data not shown)."

Reference 34 which discusses the odds ratios for BRCA1/2 is for breast cancer study not prostate as implied. Clarify the sentence.

Page 5 line 135 now states: "Effect sizes within the range known for breast cancer predisposition by pathogenic variants of *BRCA1* and *BRCA2* (ORs > 8) were observed at..."

The authors used cases from the ICPCG study. ICPCG is somewhat flawed in that they collected families whose evidence for "hereditary" falls short. Little consideration was given to degree of aggressiveness in the original collection. This can be dealt with using a table in the supplement that clarifies how cases were picked and defining groups of cases sharing characteristics. Other than early onset, do the haplotypes associate with other phenotypes? Response to therapy, outcome, aggressive disease?

ICPCG case selection is previously published (manuscript reference 27 and dbGaP data set phs000733.v1.p1 (https://www.ncbi.nlm.nih.gov/projects/gap/cgi-bin/study.cgi?study_id=phs000733.v1.p1)). HPC criteria were established by references 1 and 2 below. The criterion of three or more first- or second-degree relatives with prostate cancer defining HPC has been durably adopted by each of the ICPCG investigators. We have confidence that study HPC cases meet this criterion. HPC cases skew toward an earlier age of diagnosis, as would be expected of hereditary cancer. However, disease aggressiveness is not a recognized clinical facet of HPC. It is clinically important, but does not have an established relationship to HPC. Twin concordance studies have not yet measured the heritable component of prostate cancer severity, which is unknown. Secondary analysis of prostate cancer GWAS SNPs have observed little to also support their association with severity⁴. ICPCG investigators shared summary categorical severity data, as discussed above.

Is the 0.6 year later age at onset significant? Given that men were diagnosed during the PSA onset era—thus some were diagnosed by DRE with large tumors and others very early in their disease by rising PSA (which is never discussed), how significant can 0.6 years be?

Wilcoxon rank sum tests of association between these variants and age of diagnosis were statistically significant, with the reported *P* values (beginning page 7, line 230). The observation is unlikely to carry clinical significance. However, the concept that genetic factors may modify age of diagnosis in both directions (older and younger) is important to convey.

For the sequencing of seven cases with haplotype A, please provide more information at the onset of the section on the families and cases themselves (age at diagnosis, degree of family history, etc.), rather than sharing it piecemeal. Put a table in the supplement.

We have included an additional tab in Supplementary Table 2 providing sequenced case detail. The sample of subjects was selected to represent diverse clinical facets, each individually of unknown relationship to haplotype A. Facets considered: higher grade, stage, PSA at diagnosis, greater numbers affected in pedigree, or younger age at diagnosis. All sequenced subjects proved to be carriers of a single haplotype, corresponding to haplotype A as had been previously been assigned based upon genotyped and imputed data.

This analysis suffers from a lack of discussion regarding the likely functional impact of each variant identified as being a candidate for causal. Have the authors analyzed likely functionality using REVEL or other similar programs? Please include analysis of the four that are not discussed.

Although it is a per-base genome-wide resource, REVEL selectively omits scores for all positions between 8:127,569,634 and 128,428,112 (as of 10/10/19). CADD scores⁵ for each mutation candidate range from 0.04 to 3.47, and are not indicative of deleteriousness (page 8 line 312). Manual review of functional element data at each candidate mutation yields the noted results. The brevity of this section reflects the paucity of overlap between the candidates and current annotations. Although beyond the scope of the manuscript, annotation of 8q24 non-coding genes is incomplete (page 9 line 314); for example, reference 6 below is supported by TCGA RNA-seq data.⁶ We omitted a previously unclear sentence that had intended to state that we had failed to develop custom assays to genotype four of the mutational candidates.

Some references are out of date. The claim that analysis of hundreds of pedigrees meeting the criteria for HPC have been examined with little success cites the 1993 Carter paper. Also, the citation of one paper showing a GWAS with a significant GWAS associated at 8q24 does not do justice to the literature, as several have been published and the paper cited focuses on just one variant.

The reference was intended to orient readers to the seminal definition of HPC, at its first use in the manuscript (page 3 line 49), rather than to serve as a reference for the extensive body of linkage literature. We have added an additional reference for one of the larger linkage studies as clarification.

The introduction of 8q24 (page 3 line 66) notes that it was the very first GWAS locus observed in prostate cancer, and provides the correct citation. The following sentence states that the locus was subsequently identified by numerous GWAS of multiple cancers. We had cited the NHGRI/EMBL catalog because it is a central resource that aggregates the numerous pertinent citations for the reader. We have added citations to the prostate cancer GWAS studies with respective original discoveries at 8q24.

In the first paragraph of the discussion are the authors saying that the NFPCS study was the first to conduct a study focused on family history or hereditary cases? That is incorrect. The Hopkins study identifying the HPC1 locus was first. If the authors are saying something else, please clarify.

Page 9 beginning line 336 now clarifies: "In order to detect infrequent and strong risk variants in hereditary prostate cancer families in the setting of complexities such as incomplete penetrance and genetic heterogeneity, our study design selectively compared independent cases with a family history of the disease to controls without. To our knowledge, the NFPCS was the first conducted under this general design." This study design is distinct from the linkage design, as we used to detect HPC1.⁷

Identification of private variants in individual families are unlikely to be found by linkage, as claimed on page 9. Individual families are way too small to achieve significance and the degree of locus heterogeneity makes it impossible (see example of HOXB13).

The revised section is on page 9 line 344: "A limitation of this approach is that it would not detect rarer mutations, such as those private to a given family."

We had not intended to convey that linkage was optimal for study of HPC. Numerous challenges impede its use. Our meaning had been literal: the association design cannot find private mutations while the linkage design can. To avoid misinterpretation, we have omitted the linkage comment.

The fact that some haplotypes are inclusive, or span others is important and should be better highlighted. How much does this affect the risk analysis?

This comment/question was unclear to us. The related sections are Results page 5 beginning line 140 and Discussion page 10 beginning line 376.

One would generally expect varied causal alleles to have independent ancestral origins and to reside on distinct haplotypes, with haplotype location reflecting functional disease element location (gene, enhancer, etc). The data support the existence of more than one functional disease element at 8q24. The sentinel analysis best assesses this, though it is also reflected by disease haplotypes.

Regarding impact upon analysis, we did observe that certain risk alleles (those in light green shade on Figure 3) were found on both haplotypes A and B. The two haplotypes are distinguished by alleles in dark grey and in blue, respectively. Thus, a haplotype segment within a window spanning, for example, a light green allele and a grey allele can be assigned uniquely to haplotype A. A smaller window with a haplotype spanning only light green alleles would not distinguish between carriers of haplotypes A and B; the measure of association between that small haplotype and disease would reflect the aggregate effect of both groups of carriers. This topic is related to synthetic associations. More prominent *P* values can reflect such a scenario.

As noted on page 8 line 269, one of the sentinels is among the light green variants, shared by both haplotypes A and B.

Overall the paper does a poor job of putting these findings into the context of the published large prostate cancer GWAS, which generally are not family based, but do include family history as a variable. While those studies did not do haplotype analysis and lack the originality and

compelling results here, to not even mention them and their identification of hundreds of variants is an omission from the discussion.

Page 10 line 379 references each publication that has reported a novel 8q24 prostate cancer risk SNP. All 8q24 prostate cancer risk variants identified by prior GWAS in subjects of European ancestry were among those identified in the study. They are individually identified as such in Supplementary Table 1 on the third tab (search for the term "GWAS Variant"). We have placed these in context by assigning the minor allele for each GWAS variant to its respective haplotype (allele color code also given in the table, corresponding to that of Figure 3). We have analyzed each of the 433 replicating variants by adjusting for these GWAS variants concurrently (Supplementary Table 1 third tab). Our sentinel algorithm agnostically selected the independent risk-altering variants from the superset of all replicating variants, including previously known GWAS SNPs. The corresponding paragraph of page 7 beginning at line 263 clearly identifies each previously known GWAS variant. This study has provided considerable organization and context to the knowledge accrued by prior GWAS studies. It is an integral part of this study.

OTHER:

We corrected an allele frequency error (rs188140481 for AFR in gnomAD) in Table 4.

References within this text.

1. Carter BS, Beaty TH, Steinberg GD, Childs B, Walsh PC. Mendelian inheritance of familial prostate cancer. Proc Natl Acad Sci U S A 1992;89:3367-71.
2. Carter BS, Bova GS, Beaty TH, et al. Hereditary prostate cancer: epidemiologic and clinical features. J Urol 1993;150:797-802.
3. Berndt SI, Wang Z, Yeager M, et al. Two susceptibility loci identified for prostate cancer aggressiveness. Nat Commun 2015;6:6889.
4. Helfand BT, Roehl KA, Cooper PR, et al. Associations of prostate cancer risk variants with disease aggressiveness: results of the NCI-SPORE Genetics Working Group analysis of 18,343 cases. Hum Genet 2015;134:439-50.
5. Kircher M, Witten DM, Jain P, O'Roak BJ, Cooper GM, Shendure J. A general framework for estimating the relative pathogenicity of human genetic variants. Nat Genet 2014;46:310-5.
6. Kastler S, Honold L, Luedeke M, et al. POU5F1P1, a putative cancer susceptibility gene, is overexpressed in prostatic carcinoma. Prostate 2010;70:666-74.
7. Smith JR, Freije D, Carpten JD, et al. Major susceptibility locus for prostate cancer on chromosome 1 suggested by a genome-wide search. Science 1996;274:1371-4.

Sincerely,

Jeffrey R. Smith, MD, PhD
Associate Professor of Medicine

Reviewers' comments:

Reviewer #1 (Remarks to the Author):

The authors have comprehensively revised their manuscript responding to the reviewers' comments. However, I still do not think the description of calculation of heritability is sufficient. For example, I think that calculating heritability for a qualitative disease like prostate cancer by variance components analysis requires specifying a population lifetime prevalence of prostate cancer risk. Since that risk has varied over time due to PSA screening characteristics, the estimates of heritability are likely to vary over time and according to specifying the baseline population risk. Moreover, the method that is being applied does not seem to adjust for ascertainment of the families, which like selects for individuals who have more higher risk variants that will in turn increase the baseline probabilities of developing prostate cancer. While disentangling these effects is beyond the scope of this paper, the paper does need to be clear about what it actually did and a small sensitivity analysis would be useful. 1. What is the assumed population prevalence of prostate cancer (if that was required). 2. If you vary the population prevalence of prostate cancer within a reasonable range (by say 5 percent above and below what has been assumed in this paper) is the estimated proportion of heritability explained consistent. I would expect this is the case but it needs to be clarified. 3. If the heritability is being directly estimated from the data that are available then I expect that estimate is upwardly biased because ascertainment of the families is not being accommodated. In that case a sensitivity analysis assuming a lower baseline prevalence of prostate cancer would probably be useful to provide, if it is possible to do that. In general a bit more discussion of the foibles of heritability analysis in this manuscript would be helpful, even if that information is only presented in supplemental materials.

Reviewer #2 (Remarks to the Author):

The reviewers describe novel genetic variation and associated haplotypes that they argue alters familial risk of prostate cancer. This is a response to comments from a previous submission of the paper. The authors have done a reasonable job responding to comments. From Reviewer 1 they have redefined the terms heredity and familial which are used in a confusing way in the literature. They have clarified the case/control and allele frequency data. Their explanation of how they think about familial cases is appropriate. It is true that looking at non familial cases is always useful, but in this case it would not, as the authors correctly state, identify protective cases since some unaffected men may not have aged enough to get PCa. Other details such as the AUCreal clarification are appropriate.

With regard to reviewer two, again responses are very good. Reviewer two liked the paper, but wanted clarification on several issues, arguing, correctly that the paper was overly complex. The authors have responded appropriately for the most part. They have clarified numbers of cases and controls in response to confusion. Figure 1 is revised so that it is much easier to follow numbers of variants.

One area that can still be improved is the discussion of sentinels versus haplotypes. While it is appreciated that the authors were trying to discuss both and show the relationship between them, the presentation of a diversity of topics, at least as organized, makes it difficult to follow. The reviewer is not arguing that it is not important, simply that more signposting and presentation of rationale would be helpful.

The description of NFPCS subject selection on page 12 is helpful. It seems that the controls have no history of high PSA, prostate biopsy, and were rescreened by the study using PSA. The age differential is not an issue. The remaining issue, which can't really be fixed regards family history. While controls were men with no first- or second-degree family history, one can't correct for age (were relatives old enough to get prostate cancer), and number of eligible relatives. That should

be acknowledged.

So the assumption that early onset PC cases enriches for genetic factors is correct, but not particularly strong for prostate cancer—certainly not as strong as for, say breast cancer. See the ICPCG published data itself. Some reported associations are for older onset disease. This is in part because of the shifting diagnostic criteria for the disease in the time period in which the cases were collected, which spanned the mid 1990s. Also, not all families had 3 affected men. Some were sib pairs selected because of their age at onset. HPC cases skew to an early age at diagnosis because that is what the consortium sought to collect. However many cases within the families are older. Indeed, some families have a relatively older age at diagnosis. At the very least the time frame in which cases were collected and the strengths and limitations that result should be discussed.

The issue of cohort selection not including aggressiveness remains an inherent problem. The authors argue that twin and other studies associated heritability with onset and not aggressive disease. But many other studies show an association between genetic factors and aggressive disease. Many men with high PSA would never go on to die of their disease, and the real problem in the field is finding genes which distinguish between men dying with versus of the disease. The authors choose not to address that question, which is fine, but their results should be put in the context of leaving that question unanswered.

Other issues are well covered.

December 09, 2019

Anonymous Reviewers

Dear Reviewers,

We would like to thank you for further positive and thoughtful review of our manuscript titled "Novel 8q24 genetic variation and comprehensive haplotypes altering familial risk of prostate cancer" (NCOMMS-19-16639A).

Our responses to each of your comments are given below. The revised text has been edited to incorporate numerous facets of these comments, designated by page and line number where appropriate.

Reviewer 1.

The authors have comprehensively revised their manuscript responding to the reviewers' comments. However, I still do not think the description of calculation of heritability is sufficient. For example, I think that calculating heritability for a qualitative disease like prostate cancer by variance components analysis requires specifying a population lifetime prevalence of prostate cancer risk. Since that risk has varied over time due to PSA screening characteristics, the estimates of heritability are likely to vary over time and according to specifying the baseline population risk. Moreover, the method that is being applied does not seem to adjust for ascertainment of the families, which likely selects for individuals who have more higher-risk variants that will in turn increase the baseline probabilities of developing prostate cancer. While disentangling these effects is beyond the scope of this paper, the paper does need to be clear about what it actually did and a small sensitivity analysis would be useful.

1. What is the assumed population prevalence of prostate cancer (if that was required).

Presented results of the REML approach employ the NCI SEER estimated lifetime risk of 11.6%. Use of lifetime risk of a common cancer, vs prevalence, follows the approach of Mitchell et al.¹ and of Lu et al.² This is now noted in the Methods section on page 15, lines 627-629: "Estimates were transformed to the liability scale using the NCI SEER estimated lifetime prostate cancer risk of 11.6%.¹⁻³"

2. If you vary the population prevalence of prostate cancer within a reasonable range (by say 5 percent above and below what has been assumed in this paper) is the estimated proportion of heritability explained consistent. I would expect this is the case but it needs to be clarified.
-and-

3. If the heritability is being directly estimated from the data that are available then I expect that estimate is upwardly biased because ascertainment of the families is not being accommodated. In that case a sensitivity analysis assuming a lower baseline prevalence of prostate cancer would probably be useful to provide, if it is possible to do that.

Exploration of alternative prevalence measures did not substantively alter heritability estimates. Others have shown that only modest bias of heritability estimation on a liability scale is introduced with skew of study-represented prevalence (ascertainment).³ Methods page 15 lines 629 - 634 now states: "Results were not substantively altered by higher or lower prevalence estimates, or by adjustment for the first four principal components of genetic ancestry. For example, even assuming a lifetime risk of 0.80 for men from HPC pedigrees,⁴ the heritability explained by the 433 concordantly significant variants among combined ICPCG and NFPCS HPC subjects was 10.8% (SE = 0.027); under further adjustment for genetic ancestry, it was 10.6% (SE 0.028)." These values are similar to the value of Results page 7 line 260 (9.1%).

In general, a bit more discussion of the foibles of heritability analysis in this manuscript would be helpful, even if that information is only presented in supplemental materials.

The restricted maximum likelihood approach for estimating heritability is widely used and a brief analysis of 8q24 is presented on Results page 7, lines 255-264. As noted by reviewers, the estimate is in line with previously published, independent estimates. We have added the following to Methods page 16, lines 634-638: "Limitations by this approach are discussed in the published literature.^{3,5-9} Factors that could influence accuracy include study power, trait misspecification, genotype error or missingness, poor representation of causal alleles by LD, cryptic relatedness, or a disease architecture of causal alleles departing from an additive model."

Reviewer 2.

The reviewers describe novel genetic variation and associated haplotypes that they argue alters familial risk of prostate cancer. This is a response to comments from a previous submission of the paper. The authors have done a reasonable job responding to comments. From Reviewer 1 they have redefined the terms heredity and familial which are used in a confusing way in the literature. They have clarified the case/control and allele frequency data. Their explanation of how they think about familial cases is appropriate. It is true that looking at non-familial cases is always useful, but in this case it would not, as the authors correctly state, identify protective cases since some unaffected men may not have aged enough to get PCa. Other details such as the AUCreal clarification are appropriate.

With regard to reviewer two, again responses are very good. Reviewer two liked the paper, but wanted clarification on several issues, arguing, correctly that the paper was overly complex. The authors have responded appropriately for the most part. They have clarified numbers of cases and controls in response to confusion. Figure 1 is revised so that it is much easier to follow numbers of variants.

One area that can still be improved is the discussion of sentinels versus haplotypes. While it is appreciated that the authors were trying to discuss both and show the relationship between them, the presentation of a diversity of topics, at least as organized, makes it difficult to follow. The reviewer is not arguing that it is not important, simply that more signposting and presentation of rationale would be helpful.

We have made the following edits to better signpost the presentation, with respect to haplotypes and their relationship to sentinels:

Introduction page 4, line 85-88: "In order to resolve the distinct underlying risk-altering signals, we reconstruct ancestral risk-altering haplotypes, and introduce an algorithm systematically employing LD patterns to identify the individual variants that best detect the risk-altering signal of each of these haplotypes."

Results page 5, line 142-149: "Disease-associated alleles are inherited in the context of ancestral haplotypes. Alleles that are unique to (that "mark") the DNA segment on which a causal mutation is introduced will evidence association with disease. Recombination can diminish the correlation between a causal mutation and these alleles with time. We sought to understand how the associated alleles are transmitted as haplotypes, shedding light upon complex correlations. As discussed further below, we also identified which variant best detects the risk-altering signal(s) carried by each of these haplotypes."

Results page 8, line 279-281: "We applied the algorithm to systematically identify sentinels among those at genome-wide significance, and assessed how they were organized relative to the risk-altering haplotypes identified above."

Discussion page 10, line 395-397: "We delineated each ancestral haplotype associated with disease, and we used linkage disequilibrium patterns to identify the variants that best detect independent risk signals carried by the haplotypes."

The description of NFPCS subject selection on page 13 is helpful. It seems that the controls have no history of high PSA, prostate biopsy, and were rescreened by the study using PSA. The age differential is not an issue. The remaining issue, which can't really be fixed regards family history. While controls were men with no first- or second-degree family history, one can't correct for age (were relatives old enough to get prostate cancer), and number of eligible relatives. That should be acknowledged.

We now acknowledge the NFPCS study population control characteristic, on Methods page 13, lines 522-524: "Controls were unselected for pedigree structure (e.g. number of at-risk relatives or their ages, though the mean number of brothers of cases was 1.7 and of controls was 1.8)."

So the assumption that early onset PC cases enriches for genetic factors is correct, but not particularly strong for prostate cancer—certainly not as strong as for, say breast cancer. See the ICPCG published data itself. Some reported associations are for older onset disease. This is in part because of the shifting diagnostic criteria for the disease in the time period in which the cases were collected, which spanned the mid 1990s. Also, not all families had 3 affected men. Some were sib pairs selected because of their age at onset. HPC cases skew to an early age at diagnosis because that is what the consortium sought to collect. However, many cases within the families are older. Indeed, some families have a relatively older age at diagnosis. At the very least the time frame in which cases were collected and the strengths and limitations that result should be discussed.

The principal criterion for HPC case subject ascertainment was the number of affected men within a pedigree. This was true of both NFPCS and ICPCG HPC pedigrees evaluated in this study. While an isolated pair of affected men diagnosed under age 55 within a pedigree would also meet early criteria for HPC,¹⁰ the data set of our study did not include such pedigrees. dbGaP meta data of the ICPCG state: "Selection of Cases: Ascertainment of 1 case from each high-risk pedigree that meets the ascertainment criteria: Caucasian (European), pedigree average age at diagnosis ≤ 75 years and 3 or more related cases." This would exclude an isolated affected sib pair, though we too recall early-onset sib pairs ascertained by the ICPCG. These were not included and thus were not used in our investigation. Our presentation of the ICPCG study population is brief (Methods page 14, beginning

line 550) given extensive prior publications by the ICPCG and its individual component study groups, and given the corresponding dbGaP record compiled by ICPCG investigators.

The investigated HPC cases do have a young age of diagnosis relative to the national mean (66 yrs), presented in Table 1. For NFPCS HPC cases, age at diagnosis was not among inclusion/exclusion criteria (Methods page 13, lines 509-513) and the proband was genotyped, regardless of this age. The age distribution of NFPCS and ICPCG HPC cases is similar, even though the latter data set describes preferential genotyping of a case to represent a given pedigree as one of either aggressive disease or an early age of diagnosis (not necessarily the proband). Only 2.6% of the investigated ICPCG HPC cases and 3.6% of the investigated NFPCS HPC cases were diagnosed at an age ≥ 75 yrs. Certainly older onset cases exist within these pedigrees, other than the lone investigated case of each pedigree. Genetic and phenotypic heterogeneity within and across HPC pedigrees exists, and is an intentional undercurrent of our presentation (Introduction, first paragraph; Results page 6, paragraph beginning line 214; Discussion page 9 beginning line 356; Methods page 12, beginning line 491). A case-control design does not enable us to address within-pedigree heterogeneity. Even so, Results page 6, paragraph beginning line 214 presents a small set of NFPCS pedigrees, within which we observed that only some cases within a given HPC pedigree have inherited genetic variants carrying risk of early-onset disease.

Discussion page 10, lines 376-380, now highlights potential limitations due to heterogeneity of phenotype and of causal alleles or loci: "Potential phenotypic heterogeneity is another limitation for any such study. The investigated cases represent a spectrum of disease aggressiveness, potential pathological subtypes, and age of diagnosis. Reducing phenotypic heterogeneity could constrain genetic heterogeneity to aid investigation, though with the competing consideration of fewer subjects in a given stratum."

The issue of cohort selection not including aggressiveness remains an inherent problem. The authors argue that twin and other studies associated heritability with onset and not aggressive disease. But many other studies show an association between genetic factors and aggressive disease. Many men with high PSA would never go on to die of their disease, and the real problem in the field is finding genes which distinguish between men dying with versus of the disease. The authors choose not to address that question, which is fine, but their results should be put in the context of leaving that question unanswered. Other issues are well covered.

HPC families display a breadth of disease aggressiveness, even within a family. These families necessarily receive current standard of care, which does present inherent challenges (such as screen-detected vs clinically-detected prostate cancer). Observational studies have inherent limitations. The revised text now notes this on Methods page 12, lines 491-495: "Cases of HPC pedigrees recruited for study include both PSA screen-detected and clinically-detected disease; potential lack of uniformity of screening standard of care across recruitment sites and over time presents an inherent limitation for any such observational study. Phenotypic heterogeneity is an existing feature of the disease, complicating its study."

Aggressiveness data accompanied each ascertained HPC case to enable tests of association with 8q24 genetic variants. We have revised the manuscript to evaluate effects of the identified 8q24 variants as modifiers of disease severity. We had not previously performed this analysis on the final full data set. We expanded the Results section of page 7, line 223 to include evaluation of the identified risk-altering variants as modifiers of age at diagnosis as well as aggressiveness. The brief paragraph beginning at line 240 presents the observation that protective haplotype F is specifically enriched among cases with insignificant disease relative to cases with either moderate or aggressive disease. A corresponding bullet was also added to the Figure 1 summary (lower right).

Discussion page 10, lines 382-386, now summarizes: "We observed 8q24 haplotypes that significantly modified both age of diagnosis and aggressiveness. Intriguingly, haplotypes that were relatively

protective in effect (in excess among controls relative to cases) were also associated with a later age at diagnosis and with less severe disease among cases."

OTHER:

We corrected an error in Table 1, the percentage of controls in the 56-65 year age interval.

References

1. Mitchell JS, *et al.* Implementation of genome-wide complex trait analysis to quantify the heritability in multiple myeloma. *Sci Rep* **5**, 12473 (2015).
2. Lu Y, *et al.* Most common 'sporadic' cancers have a significant germline genetic component. *Hum Mol Genet* **23**, 6112-6118 (2014).
3. Lee SH, Wray NR, Goddard ME, Visscher PM. Estimating missing heritability for disease from genome-wide association studies. *Am J Hum Genet* **88**, 294-305 (2011).
4. Bratt O, *et al.* Family History and Probability of Prostate Cancer, Differentiated by Risk Category: A Nationwide Population-Based Study. *J Natl Cancer Inst* **108**, (2016).
5. de Los Campos G, Sorensen D, Gianola D. Genomic heritability: what is it? *PLoS Genet* **11**, e1005048 (2015).
6. Krishna Kumar S, Feldman MW, Rehkopf DH, Tuljapurkar S. Limitations of GCTA as a solution to the missing heritability problem. *Proc Natl Acad Sci U S A* **113**, E61-70 (2016).
7. Lee JJ, Chow CC. Conditions for the validity of SNP-based heritability estimation. *Hum Genet* **133**, 1011-1022 (2014).
8. Steinsaltz D, Dahl A, Wachter KW. Statistical properties of simple random-effects models for genetic heritability. *Electron J Stat* **12**, 321-356 (2018).
9. Yang J, *et al.* GCTA-GREML accounts for linkage disequilibrium when estimating genetic variance from genome-wide SNPs. *Proc Natl Acad Sci U S A* **113**, E4579-4580 (2016).
10. Carter BS, *et al.* Hereditary prostate cancer: epidemiologic and clinical features. *J Urol* **150**, 797-802 (1993).

Sincerely,

Jeffrey R. Smith, MD, PhD
Associate Professor of Medicine

REVIEWERS' COMMENTS:

Reviewer #1 (Remarks to the Author):

The authors appear to have responded to all the concerns raised by me and the other reviewer. That said, the authors did not provide a marked up version of their revision and that makes it quite hard for me to evaluate if all the comments for reviewer 2 were addressed. I think this lack of responsiveness in the review process is annoying but since it does not affect my review will suppress annoyance.

Reviewer #2 (Remarks to the Author):

The authors have done a good job responding to comments. Paragraph one of the discussion highlights much better the limitations of the various study populations and the problems with a combined data set. Issues related to methods are clarified and justified. The paper makes several unique contributions outside of the chromosome 8 locus examination in terms of methods, particularly in selecting variations of potential interest in associated haplotype blocks. The paper can still be improved by not only relating the various limitations in the study designs, particularly regarding how sample selection in the original collections were done, but by going one step further and stating how that can affect the results and interpretation so results in a more detailed way.

February 4, 2020

Anonymous Reviewers

Dear Reviewers,

We would like to thank you for further positive and thoughtful review of our manuscript titled "8q24 genetic variation and comprehensive haplotypes altering familial risk of prostate cancer" (NCOMMS-19-16639B).

Our responses to each of your comments are given below. The revised text has been edited to incorporate the request by Reviewer 2 (as well as requests of the editors) with tracked changes.

Reviewer #1:

The authors appear to have responded to all the concerns raised by me and the other reviewer. That said, the authors did not provide a marked-up version of their revision and that makes it quite hard for me to evaluate if all the comments for reviewer 2 were addressed. I think this lack of responsiveness in the review process is annoying but since it does not affect my review will suppress annoyance.

Our sincere apologies. The current revision is provided with tracked changes.

Reviewer #2:

The authors have done a good job responding to comments. Paragraph one of the discussion highlights much better the limitations of the various study populations and the problems with a combined data set. Issues related to methods are clarified and justified. The paper makes several unique contributions outside of the chromosome 8 locus examination in terms of methods, particularly in selecting variations of potential interest in associated haplotype blocks. The paper can still be improved by not only relating the various limitations in the study designs, particularly regarding how sample selection in the original collections were done, but by going one step further and stating how that can affect the results and interpretation so results in a more detailed way.

We recognize numerous potential limitations of these observational case-control studies. We have expanded paragraph one of the Discussion (page 10) to highlight some of the most pertinent aspects that may affect the results of this study. Our ability to further expand upon study limitations and their potential consequences was constrained by a separate word count reduction request for the Introduction, Results, and Discussion, and by additional requested revisions.

Sincerely,

Jeffrey R. Smith, MD, PhD
Associate Professor of Medicine